# Improving Video Generation with Human Feedback

Jie Liu[1,3,5*]  Gongye Liu[2,3*]  Jiajun Liang[3†]  Ziyang Yuan[2,3]  Xiaokun Liu[3]  Mingwu Zheng[3]
Xiele Wu[3,4]  Qiulin Wang[3]  Menghan Xia[3]  Xintao Wang[3]  Xiaohong Liu[4]  Fei Yang[3]
Pengfei Wan[3]  Di Zhang[3]  Kun Gai[3]  Yujiu Yang[2✉]  Wanli Ouyang[1,5]

[1]MMLab, CUHK      [2]Tsinghua University      [3]Kling Team, Kuaishou Technology
[4]Shanghai Jiao Tong University      [5]Shanghai AI Laboratory

[*]Equal contribution      [†]Project Leader      [✉] Corresponding author
Project page: https://gongyeliu.github.io/videoalign/

## Abstract

Video generation has achieved significant advances through rectified flow techniques, but issues like unsmooth motion and misalignment between videos and prompts persist. In this work, we develop a systematic pipeline that harnesses human feedback to mitigate these problems and refine the video generation model. Specifically, we begin by constructing a large-scale human preference dataset focused on modern video generation models, incorporating pairwise annotations across multi-dimensions. We then introduce VideoReward, a multi-dimensional video reward model, and examine how annotations and various design choices impact its rewarding efficacy. From a unified reinforcement learning perspective aimed at maximizing reward with KL regularization, we introduce three alignment algorithms for flow-based models. These include two training-time strategies: direct preference optimization for flow (Flow-DPO) and reward weighted regression for flow (Flow-RWR), and an inference-time technique, Flow-NRG, which applies reward guidance directly to noisy videos. Experimental results indicate that VideoReward significantly outperforms existing reward models, and Flow-DPO demonstrates superior performance compared to both Flow-RWR and supervised fine-tuning methods. Additionally, Flow-NRG lets users assign custom weights to multiple objectives during inference, meeting personalized video quality needs.

## 1   Introduction

Advancements in video generation have led to powerful models [57, 33, 32, 5] that produce realistic details and coherent motion. Despite this, current systems still face challenges like unstable motion, imperfect text-video alignment and insufficient alignment with human preferences [84]. In language model and image generation, reinforcement learning from human feedback (RLHF) [52, 90, 71] has proven effective in improving response quality and aligning models with user expectations.

However, applying RLHF to video generation is still remains in its infancy. A major obstacle is the lack of a reliable reward signal. Existing preference datasets [19, 73, 47, 76] were collected on earlier T2V models that produced short, low-resolution clips. Reward models trained on such data may miss fine spatial detail and long-range dynamics, while over-penalising glitches that current T2V models already suppress. In addition, the design space of VLM-based reward models remains under-explored, leading to sub-optimal annotation paradigms, reward hacking issues, and entangled multi-attribute scores. The resulting supervision is therefore noisy, biased, and easily exploited during RLHF.

A second challenge arises from the internal mechanisms of cutting-edge video generation models. Many modern systems employ rectified flow [48, 45], predicting velocity rather than noise. Recent studies [73, 85, 47, 76] have tested DPO [61, 71] and RWR [54, 36, 17] on diffusion-based video

39th Conference on Neural Information Processing Systems (NeurIPS 2025).

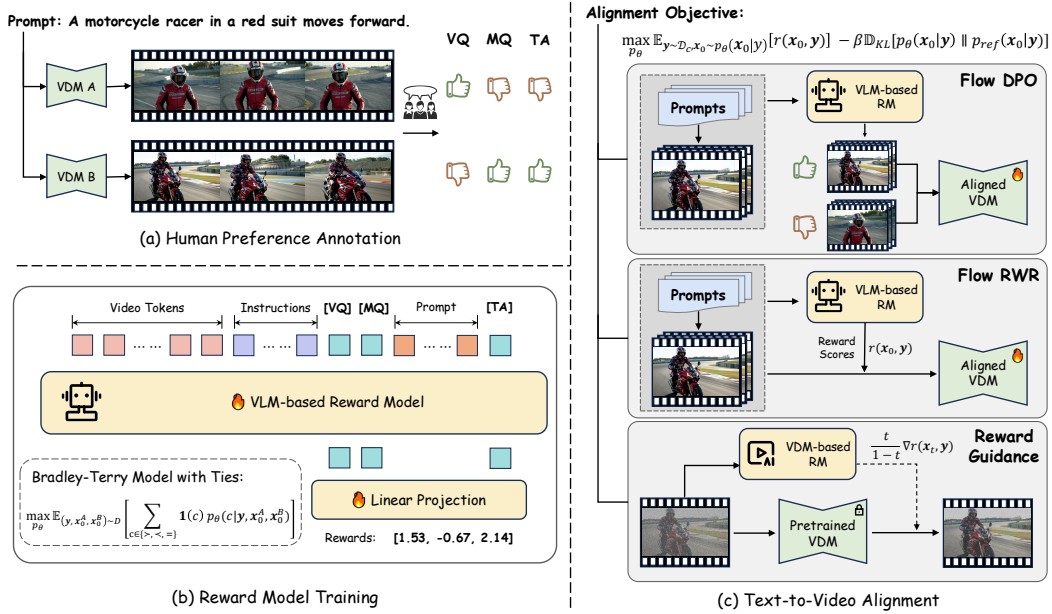

Figure 1: **Overview of Video Alignment Framework. (a) Human Preference Annotation**. We construct 182k prompt-video triplets, each annotated on Visual Quality (VQ), Motion Quality (MQ), and Text Alignment (TA). **(b) Reward Modeling**. A VLM-based reward model is trained under the Bradley-Terry-Model-with-Ties formulation. **(c) Video Alignment**. From a unified RL perspective, we introduce three alignment algorithms for flow-based video generation: Flow-DPO, Flow-RWR, and Reward Guidance (Flow-NRG), and provide a systematic comparison.

generation approaches. However, adapting existing alignment methods to flow-based models introduces new questions. A recent attempts [13] for flow matching based DPO even degrade quality compared with the unaligned baseline.

To address these challenges, we present a comprehensive investigation into aligning advanced flow-based video generation models with human preferences, as shown in Fig. 1. We first collect 16k high-quality prompts and render them with 12 representative T2V models, producing 182k annotated examples across three key dimensions: Visual Quality (VQ), Motion Quality (MQ), and Text Alignment (TA). We then develop a multi-dimensional video reward model, systematically analyzing how different annotations and design choices affect its performance. These dimensions can be aggregated into a total reward that reflects the overall preferences of humans.

Armed with a reliable reward model, we revisit RLHF algorithms for rectified flow. From a unified reinforcement learning perspective that maximizes reward with KL regularization, we derive two training-time strategies—Flow-DPO and Flow-RWR, and an inference-time technique, Flow-NRG. We discover that a simple extension of Diffusion-DPO performs poorly because its timestep-dependent KL term pushes the policy to overfit the objective at higher noise levels, making the model vulnerable to reward hacking. In contrast, our Flow-DPO removes this time-dependent term and keeps the KL weight constant, showing robust performance across all dimensions. Flow-NRG is an efficient inference-time alignment algorithm that applies reward guidance directly to noisy videos during the denoising process. It allows users to apply arbitrary weightings to multiple alignment objectives during inference, eliminating the need for extensive retraining to meet personalized user requirements.

Our contributions can be summarized as follows:

- **Large-Scale Preference Dataset**: We create a 182k-sized, multi-dimensional, human-labeled video generation preference dataset from 12 modern video generation models.

- **Multi-Dimensional Reward Modeling**: We propose and systematically study a multi-dimensional video reward model, investigating how different design decisions influence its rewarding efficacy.

- **VideoGen-RewardBench**: We create a benchmark for modern reward models by annotating prompt-video pairs from VideoGen-Eval. This dataset, consisting of diverse prompts and videos generated by modern VDMs, results in 26.5k annotated video pairs with preference labels.

- **Flow-Based Alignment**: From a unified RL perspective, we introduce two training-time alignment strategies (Flow-DPO, Flow-RWR) and one inference-time technique (Flow-NRG). Experiments show that Flow-DPO with a fixed KL term outperforms other methods. And Flow-NRG allows custom weightings of multiple alignment objectives during inference.

## 2 Related Work

Reward modeling [75, 77, 31, 86, 41] trains CLIP-based models on human preference datasets, while newer approaches use VLMs with regression heads to predict multi-dimensional scores. Learning paradigms include point-wise regression [19, 76] and pair-wise comparison via Bradley-Terry loss. However, most video reward models focus on short, low-quality videos from the pre-Sora era [51] and lack rigorous evaluation of design choices. We address these limitations by targeting modern video generation and exploring broader reward modeling strategies. For alignment, image generation has adopted RLHF-style techniques such as reward backpropagation [58, 77], RWR [36], DPO [61, 71], PPO [2, 16], and training-free methods [80]. Concurrent efforts [17, 47, 76, 85, 73] extend DPO/RWR to diffusion-based video models using old generation models or image-level rewards. We build on this by extending DPO to flow-based video generation, proposing Flow-DPO. More comprehensive discussions of related work can be found in Appendix A.

## 3 VideoReward

Robust RLHF begins with a reward model that faithfully mirrors human preferences, yet existing efforts are limited in two key respects: (1) **Data**: existing video-preference datasets were curated for earlier T2V models and mismatch with what users prefer in modern video generation models; (2) **Model design**: the key technique choices for VLM-based reward models remain largely uncharted. Fig. 1 (a), (b) summarises our solution. We first build a large-scale preference dataset collected with state-of-the-art T2V models; then perform a systematic study of reward-modeling design.

### 3.1 Human Preference Data Collection

Existing human preference datasets for video generation [49, 26, 19, 73, 76] were primarily built on early, low-resolution T2V models that produced short, artifact-laden clips. As VDMs continue to evolve, modern T2V models, however, generate longer, higher-fidelity videos with smoother motion. Consequently, legacy datasets no longer accurately reflect what users prefer today. Reward models trained on such collections may miss fine spatial details and long-range dynamics while over-weighting temporal glitches already mitigated by current models. To bridge this gap, we develop a new preference dataset expressly for state-of-the-art VDMs.

Table 1: Statistics of the collected training dataset. We utilize 12 T2V models to generate 108k videos from 16k unique prompts, resulting in 182k annotated triplets. Each triplet consists of a prompt paired with two videos, and corresponding preference annotations.

|  | T2V Model | Date | #Videos | #Anno Triplets | Resolution | Duration |
|---|---|---|---|---|---|---|
| ***Pre-Sora-Era Models*** | Gen2 [63] | 23.06 | 6k | 13k | $768 \times 1408$ | 4s |
|  | SVD [3] | 23.11 | 6k | 13k | $576 \times 1024$ | 4s |
|  | Pika 1.0 [34] | 23.12 | 6k | 13k | $720 \times 1280$ | 3s |
|  | Vega [70] | 23.12 | 6k | 13k | $576 \times 1024$ | 4s |
|  | PixVerse v1 [56] | 24.01 | 6k | 13k | $768 \times 1408$ | 4s |
|  | HiDream [21] | 24.01 | 0.3k | 0.3k | $768 \times 1344$ | 5s |
| ***Modern Models*** | Dreamina [6] | 24.03 | 16k | 68k | $720 \times 1280$ | 6s |
|  | Luma [50] | 24.06 | 16k | 57k | $752 \times 1360$ | 5s |
|  | Gen3 [64] | 24.06 | 16k | 55k | $768 \times 1280$ | 5s |
|  | Kling 1.0 [33] | 24.06 | 6k | 33k | $384 \times 672$ | 5s |
|  | PixVerse v2 [56] | 24.07 | 16k | 58k | $576 \times 1024$ | 5s |
|  | Kling 1.5 [33] | 24.09 | 7k | 28k | $704 \times 1280$ | 5s |

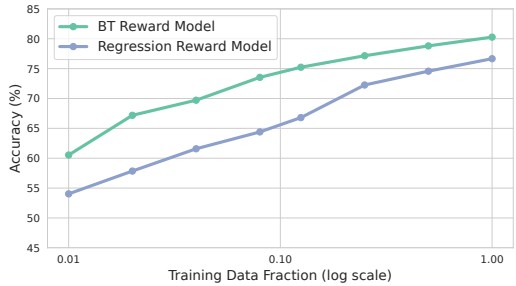
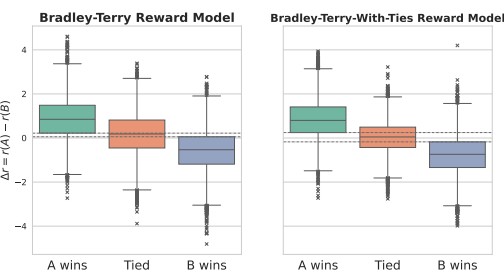

Figure 2: **BT vs. Regression.** Accuracy curves across log-scaled data fractions.

Figure 3: **BT vs. BTT.** Score-difference distributions ($\Delta r$) for BT (left) and BTT (right).

**Prompt Collection and Video Generation.** We collect diverse prompts from the Internet, categorize them into 8 meta-categories—*animal, architecture, food, people, plants, scenes, vehicles, objects*—and expand them with GPT-4o. After removing repetitive, irrelevant, or unsafe entries, we refine them with our in-house prompt rewriter, yielding **16000 high-quality prompts**. 12 T2V models of varying capabilities then render these prompts into **108k videos**, which we pair to form **182k triplets**, each comprising a prompt and two corresponding videos from distinct VDMs. Comprehensive dataset statistics are provided in Tab. 1 and Fig. 8.

**Multi-dimensional Annotation.** Professional annotators were hired to view each triplet and record pairwise preferences(*A wins / Ties / B wins*) separately for *Visual Quality (VQ)*, *Motion Quality (MQ)*, and *Text Alignment (TA)*, producing a three-label vector per triplet. The same annotators also assign 1–5 Likert scores to each individual video, enabling later studies that compare pointwise and pairwise supervision. We reserve **13 000** triplets whose prompts never appear in training as a validation set. Detailed annotation protocols are provided in Appendix G.

## 3.2 Reward Modeling

Prior works [19, 73, 76] on VLM-based reward models have proved effective for both evaluation [75, 19] and optimization [36, 71, 59]. However, their core design choices remain insufficiently explored, and current methods still suffer from issues like annotation paradigms, reward hacking and entangled multi-attribute signals. We select the lightweight Qwen2-VL-2B [72] as our backbone, and systematically examine three key designs and demonstrate how each yield a cleaner, more reliable reward signal for RLHF. We also conduct a comprehensive ablation study on the evaluation benchmark in Table 6 of Appendix E to further validate the key design choices.

**Score Regression *v.s.* Bradley-Terry.** We first investigate two reward learning paradigms: the *Bradley-Terry* (BT) model [4] and pointwise score regression. The BT model formulates preference learning as a probabilistic ranking task. Given a prompt $y$ and paired videos $(x_0^w, x_0^l)$, it optimizes $\mathcal{L}_{BT} = -\mathbb{E}\left[\log\left(\sigma\left(r(x_0^w, y) - r(x_0^l, y)\right)\right)\right]$, where the expectation is taken over $(y, x_0^w, x_0^l) \sim \mathcal{D}$.

In contrast, score regression directly predicts a scalar quality score $z \in \mathbb{R}$ using the MSE loss: $\mathcal{L}_{reg} = \mathbb{E}\left[\|r(x_0, y) - z\|^2\right]$, where the expectation is taken over $(y, x_0, z) \sim \mathcal{D}$. Since our training dataset includes both pointwise scores and pairwise preferences from the same annotators, we can directly compare between the two annotation paradigms. We train both types of reward models on increasing subsets of the training set and report the best validation accuracy averaged over VQ, MQ, and TA. Fig. 2 presents these results.

As the dataset grows, both the BT and regression-style models improve in accuracy, while the BT model remains consistently superior. This advantage stems from the nature of pairwise annotations, which capture subtle relative distinctions more effectively. Even when two videos receive identical pointwise scores, annotators can still differentiate subtle quality differences.

**Ties Matters.** While vanilla BT model is widely used to capture human preferences from chosen-rejected pairs, the importance of tie annotations is often overlooked. Inspired by recent work [46], we adopt *Bradley-Terry model with ties* (BTT) [62], an extension that accounts for tied preferences.

Formally, BTT defines a tripartite preference distribution:

$$P_\theta(c|\boldsymbol{y}, \boldsymbol{x}_0^A, \boldsymbol{x}_0^B) = \begin{cases} \dfrac{(\theta^2 - 1)\,\exp(r_A)\exp(r_B)}{(\exp(r_A) + \theta\exp(r_B))\,(\theta\exp(r_A) + \exp(r_B))} & \text{Tie } (x_0^A = x_0^B), \\[2ex] \dfrac{\exp(r_A)}{\exp(r_B) + \theta\exp(r_A)} & \text{A preferred } (x_0^A \succ x_0^B), \\[2ex] \dfrac{\exp(r_B)}{\theta\exp(r_A) + \exp(r_B)} & \text{B preferred } (x_0^B \succ x_0^A). \end{cases} \tag{1}$$

where $c$ denotes preference choice, $\theta > 1$ controls the tendency toward ties, with a larger $\theta$ increasing tie probability. We empirically set $\theta = 5.0$ and train the BTT model by minimizing negative log-likelihood:

$$\mathcal{L}_{\text{BTT}} = -\mathbb{E}_{(\boldsymbol{y}, \boldsymbol{x}_0^A, \boldsymbol{x}_0^B) \sim \mathcal{D}} \Big[ \sum_{c \in \{\succ, \prec, =\}} \mathbb{1}(c) \log P_\theta(c|\boldsymbol{y}, \boldsymbol{x}_0^A, \boldsymbol{x}_0^B) \Big] \tag{2}$$

We train BT and BTT reward models under identical settings and visualize $\Delta r = r(\boldsymbol{x}_0^A, \boldsymbol{y}) - r(\boldsymbol{x}_0^B, \boldsymbol{y})$ on the validation set (Fig. 3). Although the BT model handles chosen/rejected pairs well, it struggles to handle ties—often assigning sizeable $\Delta r$ to many tie pairs, conflating them with clear preferences. By contrast, BTT learns a flexible decision boundary that clusters ties near zero while preserving large margins for decisive wins and losses, yielding more reliable feedback for downstream RLHF.

**Token Positioning.** A common approach in LLM / MLLM-based reward modeling [52, 69, 19] attaches a linear projection head to the final token to predict multi-dimensional scores. This method forcing prompt-independent and prompt-dependent cues into one vector, causing *context leakage*: the same video can receive different visual-quality scores when paired with different prompts. We eliminate this entanglement with a simple token-positioning strategy. As shown in Fig.1(b), two context-agnostic tokens, [VQ], [MQ], are inserted immediately after the video and before the prompt, so they can attend only to visual content. A context-aware token, [TA], is placed after the full prompt, allowing it to attend to both the video and the text. The final-layer embeddings of these tokens are then mapped to dimension-specific scores via a shared linear layer. This design removes context leakage, stabilizes visual and motion assessments, and maintains parameter efficiency. The full input template is provided in Appendix K.

# 4 Video Alignment

With a high-fidelity reward model established, we next introduce three alignment methods for flow-based generation models under a unified RL objective: two training-time algorithms—**Flow-DPO** and **Flow-RWR**, and one inference-time guidance technique, **Flow-NRG** (Fig. 1 (c)).

## 4.1 Preliminaries

**Rectified Flow.** Let $\boldsymbol{x}_0 \sim q(\boldsymbol{x}_0)$ denote a data sample from the real data distribution, and $\boldsymbol{x}_1 \sim p(\boldsymbol{x}_1)$ denote a noise sample, where $\boldsymbol{x}_0, \boldsymbol{x}_1 \in \mathbb{R}^d$. Recent advanced image [15] and video generation models [57, 32] adopt the Rectified Flow [48], which defines the "noisy" data $\boldsymbol{x}_t$ as $\boldsymbol{x}_t = (1 - t)\,\boldsymbol{x}_0 + t\,\boldsymbol{x}_1$, where $t \in [0, 1]$. Then we can train a transformer model to regress the velocity field $\boldsymbol{v}_\theta(\boldsymbol{x}_t, t)$ by minimizing the Flow Matching objective [45, 48]:

$$\mathcal{L}(\theta) = \mathbb{E}_{t,\,\boldsymbol{x}_0 \sim q(\boldsymbol{x}_0),\,\boldsymbol{x}_1 \sim p(\boldsymbol{x}_1)} \big[\, \|\boldsymbol{v} - \boldsymbol{v}_\theta(\boldsymbol{x}_t, t)\|^2 \,\big],$$

where the target velocity field is $\boldsymbol{v} = \boldsymbol{x}_1 - \boldsymbol{x}_0$.

**RLHF.** The goal of RLHF is to learn a conditional distribution $p_\theta(\boldsymbol{x}_0 \mid \boldsymbol{y})$ that maximizes the reward $r(\boldsymbol{x}_0, \boldsymbol{y})$ while controls the KL-divergence from the reference model $p_{\text{ref}}$ via a coefficient $\beta$:

$$\max_{p_\theta} \mathbb{E}_{\boldsymbol{y} \sim \mathcal{D}_c, \boldsymbol{x}_0 \sim p_\theta(\boldsymbol{x}_0|\boldsymbol{y})} [r(\boldsymbol{x}_0, \boldsymbol{y})] - \beta\, \mathbb{D}_{\text{KL}} [p_\theta(\boldsymbol{x}_0 \mid \boldsymbol{y}) \,\|\, p_{\text{ref}}(\boldsymbol{x}_0 \mid \boldsymbol{y})]. \tag{3}$$

## 4.2 Flow-DPO

Consider a training set $\mathcal{D} = \{\boldsymbol{y}, \boldsymbol{x}_0^w, \boldsymbol{x}_0^l\}$, where $y$ is the prompt and human annotations prefer generated video $\boldsymbol{x}_0^w$ to $\boldsymbol{x}_0^l$ (i.e., $\boldsymbol{x}_0^w \succ \boldsymbol{x}_0^l$). Direct Preference Optimization (DPO) [61] aligns models with human preferences by analytically solving the RLHF objective (Eq. 3) and optimizing the policy via supervised learning. Extending this idea to diffusion models, Diffusion-DPO [71] derives a DPO-style loss under the diffusion paradigm. The resulting objective $\mathcal{L}_{\text{DD}}(\theta)$ is formulated as:

$$-\mathbb{E}\left[\log \sigma\left(-\frac{\beta}{2}\Big(\|\boldsymbol{\epsilon}^w - \boldsymbol{\epsilon}_\theta(\boldsymbol{x}_t^w, t)\|^2 - \|\boldsymbol{\epsilon}^w - \boldsymbol{\epsilon}_{\text{ref}}(\boldsymbol{x}_t^w, t)\|^2 - \big(\|\boldsymbol{\epsilon}^l - \boldsymbol{\epsilon}_\theta(\boldsymbol{x}_t^l, t)\|^2 - \|\boldsymbol{\epsilon}^l - \boldsymbol{\epsilon}_{\text{ref}}(\boldsymbol{x}_t^l, t)\|^2\big)\Big)\right)\right] \quad (4)$$

The expectation is taken over samples $\{\boldsymbol{x}_0^w, \boldsymbol{x}_0^l\} \sim \mathcal{D}$ and the noise schedule $t$. In Rectified Flow, we relate the noise vector $\boldsymbol{\epsilon}^*$ to a velocity field $\boldsymbol{v}^*$. Specifically, Lemma B.1 in Appendix B shows that

$$\|\boldsymbol{\epsilon}^* - \boldsymbol{\epsilon}_{\text{pred}}(\boldsymbol{x}_t^*, t)\|^2 = (1-t)^2 \|\boldsymbol{v}^* - \boldsymbol{v}_{\text{pred}}(x_t^*, t)\|^2, \quad (5)$$

By substituting Eq. 5 into Eq. 4, we obtain the final Flow-DPO loss $\mathcal{L}_{\text{FD}}(\theta)$:

$$-\mathbb{E}\Bigg[\log \sigma\bigg(-\frac{\beta_t}{2}\Big(\|\boldsymbol{v}^w - \boldsymbol{v}_\theta(\boldsymbol{x}_t^w, t)\|^2 - \|\boldsymbol{v}^w - \boldsymbol{v}_{\text{ref}}(\boldsymbol{x}_t^w, t)\|^2$$
$$- \big(\|\boldsymbol{v}^l - \boldsymbol{v}_\theta(\boldsymbol{x}_t^l, t)\|^2 - \|\boldsymbol{v}^l - \boldsymbol{v}_{\text{ref}}(\boldsymbol{x}_t^l, t)\|^2\big)\Big)\bigg)\Bigg] \quad (6)$$

where $\beta_t = \beta(1-t)^2$. Intuitively, minimizing $\mathcal{L}_{\text{FD}}(\theta)$ guides the predicted velocity field $\boldsymbol{v}_\theta$ closer to the target velocity $\boldsymbol{v}^w$ of the "preferred" data, while pushing it away from $\boldsymbol{v}^l$ (the "less preferred" data). The strength of this preference signal depends on the differences between the predicted errors and the corresponding reference errors. We provide Flow-DPO pseudo-code in Appendix C.

**Discussion on $\beta_t$.** The KL coefficient $\beta_t$ controls how far the learned policy is allowed to deviate from the reference model [61, 71]. A direct derivation yields the schedule $\beta_t = \beta(1-t)^2$. The penalty $\beta_t$ vanishes as $t$ approaches 1 and reaches $\beta$ at $t = 0$. *This scheduling strategy causes the model to prioritize alignment at higher noise levels.* Unlike in diffusion models, our experiments reveal that this schedule degrades alignment performance in rectified flow, leading to reward hacking and visual artifacts. Inspired by a similar observation in DDPM's [22] training objective, where discarding the weighting in denoising score matching improves sample quality, we instead adopt a constant $\beta$. This adjustment leads to more stable training and improved alignment across all reward dimensions. We provide a more detailed discussion of this in Section 5.2.

## 4.3 Flow-RWR

Drawing inspiration from the application of Reward-weighted Regression (RWR) [55] in diffusion models [36, 17], we propose a counterpart for flow-based models based on expectation-maximization [10]. Starting from the general KL-regularized reward-maximization problem in Eq. 3, prior work [61] shows that its optimal closed-form solution can be written as:

$$p_\theta(\boldsymbol{x}_0 \mid \boldsymbol{y}) = \frac{1}{Z(\boldsymbol{y})} p_{\text{ref}}(\boldsymbol{x}_0 \mid \boldsymbol{y}) \exp\left(\frac{1}{\beta} r(\boldsymbol{x}_0, \boldsymbol{y})\right), \quad (7)$$

where $Z(\boldsymbol{y}) = \sum_{\boldsymbol{x}_0} p_{\text{ref}}(\boldsymbol{x}_0 \mid \boldsymbol{y}) \exp\big(\frac{1}{\beta} r(\boldsymbol{x}_0, \boldsymbol{y})\big)$ is the partition function. Following [17], we can obtains the RWR loss:

$$\mathcal{L}_{\text{RWR}}(\theta) = \mathbb{E}_{\boldsymbol{y}, \boldsymbol{x}_0, \boldsymbol{\epsilon}, t}\big[\exp(r(\boldsymbol{x}_0, \boldsymbol{y}))\|\boldsymbol{\epsilon} - \boldsymbol{\epsilon}_\theta(\boldsymbol{x}_t, t, \boldsymbol{y})\|^2\big]. \quad (8)$$

For rectified-flow models, we formulate this as a reward-weighted velocity regression:

$$\mathcal{L}_{\text{RWR}}(\theta) = \mathbb{E}\big[\exp(r(\boldsymbol{x}_0, \boldsymbol{y}))\|\boldsymbol{v} - \boldsymbol{v}_\theta(\boldsymbol{x}_t, t, \boldsymbol{y})\|^2\big], \quad (9)$$

As in Flow-DPO, we omit the $(1-t)^2$ factor for better performance.

Table 2: Preference accuracy on GenAI-Bench and VideoGen-RewardBench. For ties-excluded accuracy, we calculate accuracy using only the data labeled as "A wins" or "B wins". For ties-included accuracy, we use the algorithm from Deutsch et al. [11], which tests various tie thresholds and selects the one that maximizes three-class accuracy. **Bold**: best performance.

| Method | GenAI-Bench | | VideoGen-RewardBench | | | | | | | |
| | Overall Accuracy | | Overall Accuracy | | VQ Accuracy | | MQ Accuracy | | TA Accuracy | |
| | w/ Ties | w/o Ties | w/ Ties | w/o Ties | w/ Ties | w/o Ties | w/ Ties | w/o Ties | w/ Ties | w/o Ties |
|---|---|---|---|---|---|---|---|---|---|---|
| Random | 33.67 | 49.84 | 41.86 | 50.30 | 47.42 | 49.86 | 59.07 | 49.64 | 37.25 | 50.40 |
| VideoScore [19] | 49.03 | 71.69 | 41.80 | 50.22 | 47.41 | 47.72 | 59.05 | 51.09 | 37.24 | 50.34 |
| LiFT [73] | 37.06 | 58.39 | 39.08 | 57.26 | 47.53 | 55.97 | 59.04 | 54.91 | 33.79 | 55.43 |
| VisionRewrd [76] | **51.56** | 72.41 | 56.77 | 67.59 | 47.43 | 59.03 | 59.03 | 60.98 | 46.56 | 61.15 |
| Ours | 49.41 | **72.89** | **61.26** | **73.59** | **59.68** | **75.66** | **66.03** | **74.70** | **53.80** | **72.20** |

## 4.4 Noisy Reward Guidance

Recall that the KL-regularized RL objective (Eq. 3) admits a closed-form solution(Eq. 7), which transform the original distribution $p_{\text{ref}}(\boldsymbol{x}_0 \mid \boldsymbol{y})$ into the new target distribution $p_\theta(\boldsymbol{x}_0 \mid \boldsymbol{y})$. Since the constants $\beta$ and $w$ can be absorbed into $r(\boldsymbol{x}_0, \boldsymbol{y})$, the closed-form solution becomes:

$$p_\theta(\boldsymbol{x}_0 \mid \boldsymbol{y}) \; \propto \; p_{\text{ref}}(\boldsymbol{x}_0 \mid \boldsymbol{y}) \left[\exp(r(\boldsymbol{x}_0, \boldsymbol{y}))\right]^w, \quad (10)$$

where $w \in \mathbb{R}$ controls the strength of the reward guidance. For rectified flow, as we proved in Appendix B.2, this reweighting can be achieved by shifting the velocity field:

$$\tilde{\boldsymbol{v}}_t(\boldsymbol{x}_t \mid \boldsymbol{y}) = \boldsymbol{v}_t(\boldsymbol{x}_t \mid \boldsymbol{y}) - w \frac{t}{1-t} \nabla r(\boldsymbol{x}_t, \boldsymbol{y}), \quad (11)$$

This modification of the marginal velocity field alters the sampling distribution to match the target form in Eq. equation 10. Since this formulation is structurally similar to classifier guidance [12, 67], we refer to it as *reward guidance*. Pseudo-code is provided in Appendix C.

**Efficient Reward on Noisy Latents.** Computing $\nabla r$ in pixel space requires back-propagating through the full VAE decoder, which is computationally expensive. To address this, we propose training a lightweight, **time-dependent reward model** $r_\theta(\cdot, t)$ directly in latent space. For each preference pair $(\boldsymbol{x}^w, \boldsymbol{x}^l)$, we apply identical noise to both videos, assuming their relative preference remains unchanged. We then adopt the Bradley–Terry loss to learn the reward function from these noised videos. Leveraging the fact that modern VDMs are already well trained on noisy latents, we can reuse a few early layers from the pretrained backbone to construct the reward model, avoiding the need for heavy retraining. We apply Eq. equation 11 at each inference step (except at $t = 1$), enabling efficient inference-time alignment in latent space.

## 5 Experiments

### 5.1 Reward Learning

**Training Setting.** We use Qwen2-VL-2B [72] as the backbone of our reward model, trained with BTT loss. Several observations were made during training. First, higher video resolution and more frames generally improved the reward model's performance. Second, using a stable sampling interval instead of a fixed frame number significantly enhanced motion quality evaluations, especially for videos of varying lengths. In practice, we sample videos at 2 fps, with a resolution of approximately $448 \times 448$ pixels while preserving the original asoect ratio. Hyperparameters are in Appendix I.

**Evaluation.** We evaluate our reward model on two benchmarks targeting different generations of T2V models: **(1) VideoGen-RewardBench**: Built upon the third-party prompt-video dataset VideoGen-Eval [84], this benchmark targets *modern T2V models*. We address the lack of human annotations in VideoGen-Eval by manually constructing 26.5k triplets and hiring annotators to provide pairwise preference labels. Annotators also assess overall video quality, serving as a universal label across all dimensions. **(2) GenAI-Bench** [29]: GenAI-Bench features short (2-seconds) videos generated by *pre-Sora-era T2V models*, enabling evaluation on earlier-generation outputs. A detailed comparison between the two benchmarks is provided in Appendix H.1. We evaluate our reward model against existing baselines, including VideoScore [19], as well as two concurrent works: LiFT [73]

Table 3: Multi-dimensional alignment with VQ:MQ:TA = 1:1:1. **Bold**: Best performance. Although Flow-DPO with a timestep-dependent $\beta$ achieves high VQ and MQ reward win rates, it exhibits significant reward hacking. In contrast, Flow-DPO with a constant $\beta$ achieves high VQ, MQ, and TA scores while avoiding reward hacking.

| Method | VBench | | | | | | VideoGen-Eval | | | TA-Hard | | |
|---|---|---|---|---|---|---|---|---|---|---|---|---|
| | Total | Quality | Sementic | VQ | MQ | TA | VQ | MQ | TA | VQ | MQ | TA |
| Pretrained | 83.19 | **84.37** | 78.46 | 50.0 | 50.0 | 50.0 | 50.0 | 50.0 | 50.0 | 50.0 | 50.0 | 50.0 |
| SFT | 82.31 | 83.13 | 79.04 | 51.28 | 65.21 | 52.84 | 61.27 | 76.13 | 46.35 | 57.75 | 76.06 | 57.75 |
| Flow-RWR | 82.27 | 83.19 | 78.59 | 51.55 | 63.9 | 53.43 | 59.05 | 69.7 | 48.35 | 61.97 | 78.87 | 55.71 |
| Flow-DPO ($\beta_t = \beta(1-t)^2$) | 80.90 | 81.52 | 78.42 | 87.78 | **82.36** | 51.02 | 88.44 | **91.23** | 28.14 | **84.29** | **83.10** | 38.03 |
| Flow-DPO | **83.41** | 84.19 | **80.26** | **93.42** | 69.08 | **75.43** | **90.95** | 81.01 | **68.26** | 77.46 | 71.43 | **73.24** |

Table 4: Single-dimensional alignment with TA. **Bold**: Best performance. Flow-DPO with a constant $\beta$ is the most effective method, achieving best performance without reward hacking.

| Method | VBench | | | | VideoGen-Eval | TA-Hard |
|---|---|---|---|---|---|---|
| | Total | Quality | Semantic | TA | TA | TA |
| Pretrained | 83.19 | **84.37** | 78.46 | 50.00 | 50.00 | 50.00 |
| SFT | 82.71 | 83.48 | 79.62 | 52.88 | 53.81 | 64.79 |
| Flow-RWR | 82.40 | 83.36 | 78.58 | 59.66 | 49.50 | 66.20 |
| Flow-DPO ($\beta_t = \beta(1-t)^2$) | 82.35 | 83.00 | 79.75 | 63.67 | 55.95 | 71.83 |
| Flow-DPO | **83.38** | 84.28 | **79.80** | **69.09** | **65.49** | **84.51** |

and VisionReward [76]. Consistent with practices in LLM evaluation [35], we use pairwise accuracy, reporting both ties-included [11] and ties-excluded accuracy. We calculate overall accuracy on GenAI-Bench and dimension-specific (VQ, MQ, TA) accuracy on VideoGen-RewardBench. Additional evaluation details are in Appendix H.2 and Appendix H.3

**Main Results.** Tab. 2 presents the pairwise accuracy across both benchmarks. **VideoScore** performs well on GenAI-Bench but fails on VideoGen-RewardBench, indicating poor generalization to modern T2V models. **LiFT** improves over VideoScore on modern videos but remains below 60% accuracy, showing limited pairwise discrimination ability. **VisionReward** demonstrates competitive performance on GenAI-Bench but underperforms on VideoGen-RewardBench, especially on visual and motion dimensions under ties-included settings. This drop stems from its difficulty in assessing the improved fidelity and motion smoothness of modern outputs. In contrast, our method **VideoReward** outperforms all other models on VideoGen-RewardBench, showcasing its strong alignment with human preferences on modern T2V generations. Moreover, despite being trained on a disjoint dataset (see Fig. 10), it still achieves comparable performance on GenAI-Bench, indicating robust generalization across different eras of T2V models. **Ablation studies are provided in Appendix E.**

## 5.2 Video Alignment

**Training Setting.** Our pretrained model $p_{\text{ref}}$ is an internal, research-purpose video generation model based on Transformer architecture [53], which is trained using rectified flow (see Appendix 7 for details). Following SD3 [15], all alignment experiments fine-tune the Transformer using LoRA [24]. For training-based alignment methods, including SFT, Flow-DPO, and Flow-RWR, we adopt Video-Reward to provide the reward signals. For reward guidance, we employ the *latent* reward model to generate rewards. For supervised fine-tuning (**SFT**), we utilize only the "chosen data". Following Rafailov et al. [61], we employ VideoReward as the ground-truth reward model to simulate human feedback and relabel our training dataset, ensuring that models optimised on these synthetic labels can be evaluated fairly by the same reward function. Hyperparameter settings are provided in the Appendix I.

**Evaluation.** We evaluate performance using both automatic and human assessments. For automatic evaluation, we measure the *win rate* (via VideoReward) and the *Vbench* score. **The win rate** is the proportion of cases where VideoReward assigns a higher reward to the aligned model than to the pretrained model. **Vbench** is a fine-grained T2V benchmark that assesses Quality and Semantic alignment. For human evaluation, each sample is reviewed by two annotators, with a third annotator resolving disagreements. Identical random seeds are used across methods for strict comparability. Our evaluation uses prompts from *Vbench*, *VideoGen-Eval*, and a new *TA-Hard* set that stresses

A cowboy rides his horse across an open plain at sunset, with warm sky colors and soft lighting on the landscape.

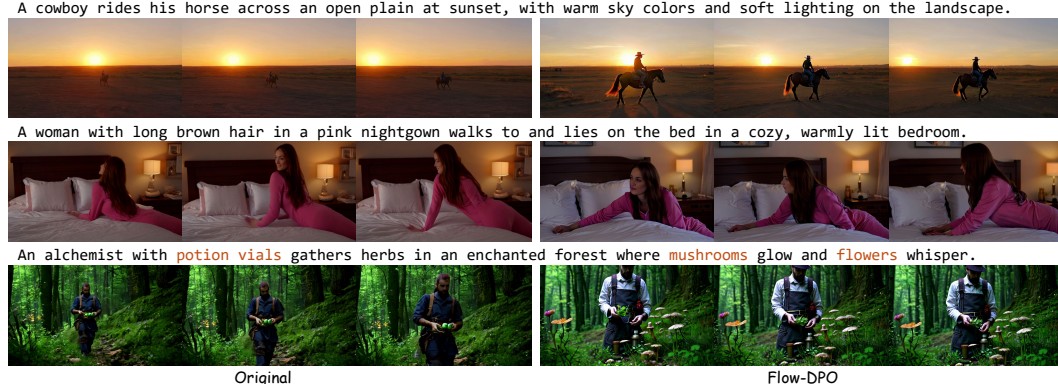

A woman with long brown hair in a pink nightgown walks to and lies on the bed in a cozy, warmly lit bedroom.

An alchemist with potion vials gathers herbs in an enchanted forest where mushrooms glow and flowers whisper.

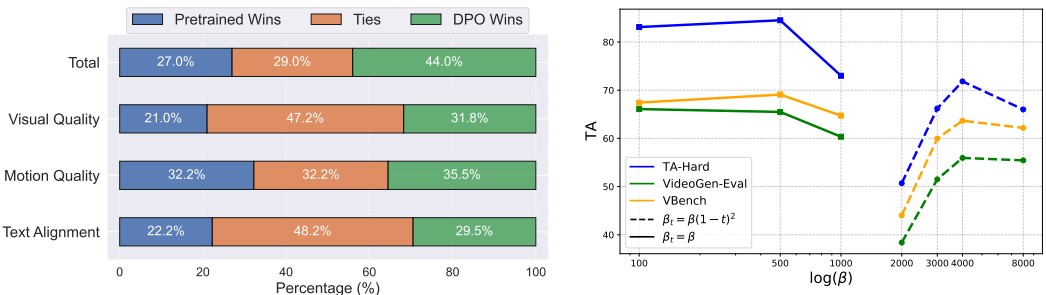

Original                                    Flow-DPO

Figure 4: Comparison of videos generated by the original model and the Flow-DPO aligned model.

Figure 5: Human evaluation of Flow-DPO aligned model *vs.* pretrained model on VideoGen-Eval.

Figure 6: Accuracy of time-dependent $\beta_t$ *v.s.* constant $\beta$ for TA.

complex semantics, since we notice that the Vbench and VideoGen-Eval prompts are relatively straightforward in terms of text alignment. Appendix L lists a subset of TA-Hard prompts.

**Multi-dimensional Alignment.** We use linear scalarization [40], $r = \frac{1}{3}(r_{\mathrm{vq}} + r_{\mathrm{mq}} + r_{\mathrm{ta}})$, to aggregate multi-dimensional preferences into a single score, and forming a dataset $D = \{(\boldsymbol{y}, \boldsymbol{x}^w, \boldsymbol{x}^l)\}$ where $r(\boldsymbol{x}^w, \boldsymbol{y}) > r(\boldsymbol{x}^l, \boldsymbol{y})$. Table 3 shows that Flow-DPO with a constant $\beta$ significantly improves over the pretrained model across all three dimensions and outperforms both SFT and Flow-RWR. In contrast, Flow-DPO with timestep-dependent $\beta$ underperforms the pretrained model on TA, despite high VQ/MQ reward win rates due to reward hacking. Meanwhile, the constant-$\beta$ variant achieves high VQ/MQ scores without such hacking issues, suggesting it learns TA more effectively. We discuss this further in Sec. 5.2. Figures 5 confirms these findings in human studies.

**Single-dimensional Alignment.** We also investigate the ability of different methods on a specific task: TA. Tab. 4 shows that Flow-DPO with a constant $\beta$ achieves best performance across all datasets.

**Reward Guidance.** We apply linear scalarization to combine dimension rewards into a weighted sum and backpropagate gradients to the noised latent with guidance strength $w = 100$. Table 5 shows that users can steer generation toward custom trade-offs by simply choosing custom weights.

Table 5: Reward guidance on VideoGen-Eval.

| VQ:MQ:TA | VQ | MQ | TA |
|---|---|---|---|
| 0:0:1 | 60.56 | 46.48 | 70.42 |
| 0.1:0.1:0.8 | 66.50 | 63.73 | 60.86 |
| 0.1:0.1:0.6 | 68.94 | 67.59 | 53.28 |
| 0.5:0.5:0 | 86.43 | 93.23 | 26.65 |

**Ablation on $\beta$.** We meticulously adapted diffusion-DPO to flow-based models, resulting in Equation 6, where $\beta_t = \beta(1-t)^2$. Figure 6 shows that under various $\beta$ values, a constant $\beta$ consistently outperforms the timestep-dependent variant. This is likely because varying $\beta$ across timesteps leads to uneven training [18], given that T2V models use shared weights for different noise levels.

# 6 Conclusion

We present a large-scale preference dataset of 182k human annotations covering visual quality, motion quality, and text alignment for modern video generation models. Building on this dataset, we introduce *VideoReward*, a multi-dimensional video reward model, and establish the *VideoGen-RewardBench* benchmark for more accurate and fair evaluations. From a unified reinforcement learning perspective, we further propose three alignment algorithms tailored to flow-based video generation, demonstrating their effectiveness in practice.

# 7 Limitations & Future Work

In our experiments, excessive training with Flow-DPO led to a significant deterioration in model quality, despite improvements in alignment across specific dimensions. To prevent this decline, we employed LoRA training. Future work can explore the simultaneous use of high-quality data for supervised learning during DPO training, aiming to preserve video quality while enhancing alignment. Additionally, our algorithms have been validated on text-to-video tasks; future work can extend the validation to other conditional generation tasks, such as image-to-video generation. Moreover, despite our efforts like incorporating tie annotations into the modeling, our VideoReward model still carries a potential risk of reward hacking, where human assessments indicate a marked decrease in video quality, yet the reward model continues to assign high scores. This issue arises because the reward function is differentiable, making it susceptible to manipulation. Future research should focus on developing more robust reward models, potentially by incorporating uncertainty estimates and increasing data augmentation. Additionally, there is potential to apply more RLHF algorithms, such as PPO, to flow-based video generation tasks.

## Acknowledgements

This work was supported by the JC STEM Lab of AI for Science and Engineering, which is funded by The Hong Kong Jockey Club Charities Trust and the Research Grants Council of Hong Kong (Project No. CUHK14213224). It was also supported by the National Natural Science Foundation of China (Grant No. 62576191).

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

# Appendix of Improving Video Generation with Human Feedback

Our Appendix consists of 8 sections. Readers can click on each section number to navigate to the corresponding section:

## A    Related Works

**Evaluation and Reward Models.**    Evaluation models and reward models play a pivotal role in aligning generative models with human preferences. Earlier approaches and benchmarks [25, 49, 26] relied on metrics like FID [20] and CLIP scores [60] to assess visual quality and semantic consistency. Recent works [75, 77, 31, 86, 41] have shifted towards utilizing human preference datasets to train CLIP-based models, enabling them to predict preference scores with improved accuracy. With the advent of large vision-language models (VLMs) [1, 72], their powerful capabilities in visual understanding and text-visual alignment make them a natural proxy for reward modeling. A common approach involves replacing the token classification head of VLMs with a regression head that predicts multi-dimensional scores for diverse evaluation tasks.

Two main learning paradigms have emerged based on the type of human annotation. The first paradigm relies on point-wise regression, where the model learns to fit annotated scores [19] or labels [76] directly. Another paradigm focuses on pair-wise comparisons [74], leveraging Bradley-Terry (BT) [4] loss or rank loss to model relative preferences, which is largely unexplored for video reward model. Beyond these methods, some works [73] also leverage the intrinsic reasoning capabilities of VLMs through VLM-as-a-judge [39, 44], where VLMs are adopted to generate preference judgments or scores in textual format through instruction tuning. Despite these promising advances, most existing video reward models primarily focus on legacy video generation models, typically from the pre-Sora [51] era, which are constrained by short video durations and relatively low quality. Furthermore, the technical choices underlying the vision reward models remain underexplored. Our work seeks to address these limitations by focusing more on modern video generation models and investigating a broader range of reward modeling strategies.

**Alignment for Image & Video Generation.** In large language models, Reinforcement Learning from Human Feedback (RLHF) improves alignment with human preferences, enhancing response quality [52, 27]. Similar methods have been applied to image generation, using reward models or human preference data to align pretrained models. Key approaches include: (1) direct backpropagation with reward signals [58, 9, 77, 59]; (2) Reward-Weighted Regression (RWR) [54, 36, 17]; (3) DPO-based policy optimization [61, 71, 42, 14, 78, 42, 83, 47, 85, 17]; (4) PPO-based policy gradients [65] [2, 16, 87, 7]; and (5) training-free alignment [80, 68, 66]. These methods have successfully aligned image generation models with human preferences, improving aesthetics and semantic consistency. They focus on improving the accuracy of rewards on noised images using reward models trained on clean images, whereas our Flow-NRG directly trains noise-aware reward models to obtain accurate gradients. Our work applies the DPO algorithm [61, 71] to flow matching in video generation. Concurrent work [13] also explores similar things in image generation. However, they reports worse performance for the DPO-aligned model compared to the unaligned one. We argue that the originally derived Flow-DPO algorithm imposes a stronger KL constraint at lower noise steps, resulting in suboptimal performance. In contrast, using a constant KL constraint significantly improves performance on certain tasks. Some prior work explores aligning video generation models using direct backpropagation with differentiable rewards [82, 38, 37, 59], often relying on image reward models [31, 75]. However, these approaches cannot be directly applied to modern T2V generation with a VLM-based video reward and large VAE decoders, as they exceed the memory limits of existing GPU setups, requiring specialized engineering techniques to handle.

**Discussion with Concurrent Video Alignment Works.** Several concurrent works also explore aligning video generation models using feedback. Furuta et al. [17] derives a unified probabilistic objective for offline RL fine-tuning of text-to-video models for DPO and RWR. VideoDPO [47] introduces a re-weighting factor on the Diffusion-DPO loss to adjust the impact of each pair, encouraging the model to learn more effectively from pairs with clearer distinctions. VisionReward [76] ensures that all dimensions of the chosen data outperform those of the rejected data, addressing the issue of confounding factors in preference data. OnlineVPO [85] presents an online DPO algorithm to tackle off-policy optimization. LIFT [73] proposes applying RWR on synthesized datasets while simultaneously performing supervised learning on real video-text datasets, as synthesized videos often suffer from low temporal consistency. All of these works use Diffusion-DPO or Diffusion-RWR and focus on aligning diffusion-based video generation models, where the videos in the preference datasets are either generated by earlier open-source models or use image reward models directly. In contrast, our work explores alignment techniques for advanced flow-based video generation. We extend Diffusion-DPO into Flow-DPO, but our derivation reveals that the parameter $\beta$ (the KL divergence constraint) in Flow-DPO is timestep-dependent, which leads to suboptimal performance on certain tasks. However, fixing $\beta$ resolves this issue. SPO [42] also assumes that the preference order between pairs of images remains consistent when adding the same noise, and constructs win-lose pairs for noised images, proposing step-aware preference optimization. Our work differs from SPO in that while SPO focuses on improving the training method DPO, our Flow-NRG specifically targets training noise-ware reward model on noised videos.

# B    Details of the Derivation

## B.1    Relation beween Velocity and Noise

**Lemma B.1.** *Let $X_0 \sim q$ be a real data sample drawn from the true data distribution and $X_1 \sim p$ be a noise sample, where $X_0, X_1 \in \mathbb{R}^d$. Define $\boldsymbol{v}_t(\boldsymbol{x}_t \mid X_0, X_1)$ to be the conditional velocity field specified by a Rectified Flow [48], and let $\boldsymbol{v}_t^{pred}(\boldsymbol{x}_t)$ be the predicted* marginal *velocity field. Then the L2 error of the noise prediction is related to the L2 error of the velocity field prediction by*

$$\|X_1 - X_1^{pred}(\boldsymbol{x}_t, t)\|^2 \;=\; (1-t)^2 \left\|\boldsymbol{v}_t(\boldsymbol{x}_t \mid X_0, X_1) - \boldsymbol{v}_t^{pred}(\boldsymbol{x}_t)\right\|^2. \tag{12}$$

*Proof.* The Rectified Flow is a time-dependent flow $\psi : [0, 1] \times \mathbb{R}^d \to \mathbb{R}^d$ for $t \in [0, 1]$, defined by

$$\psi(X_0, X_1) \;=\; (1-t)\, X_0 \;+\; t\, X_1.$$

By definition, the *marginal* velocity field $\boldsymbol{v}_t(\boldsymbol{x}_t)$ is

$$
\begin{aligned}
\boldsymbol{v}_t(\boldsymbol{x}_t) &= \mathbb{E}\big[\boldsymbol{v}_t(X_t \mid X_0, X_1) \mid X_t = \boldsymbol{x}_t\big] \\
&= \mathbb{E}\big[\dot{\psi}(X_t \mid X_0, X_1) \mid X_t = \boldsymbol{x}_t\big] \\
&= \mathbb{E}\big[X_1 - X_0 \mid X_t = \boldsymbol{x}_t\big] \\
&= \mathbb{E}\Big[\frac{X_1 - \big((1-t)\,X_0 + t\,X_1\big)}{1-t} \,\Big|\, X_t = \boldsymbol{x}_t\Big] \\
&= \mathbb{E}\Big[\frac{X_1 - \boldsymbol{x}_t}{1-t} \,\Big|\, X_t = \boldsymbol{x}_t\Big] \\
&= \frac{\mathbb{E}[\,X_1 \mid X_t = \boldsymbol{x}_t\,] - \boldsymbol{x}_t}{1-t}.
\end{aligned}
\tag{13}
$$

Meanwhile, the *conditional* velocity field $\boldsymbol{v}_t(\boldsymbol{x}_t \mid X_0, X_1)$ is given by

$$
\boldsymbol{v}_t(\boldsymbol{x}_t \mid X_0, X_1) \;=\; \frac{X_1 - \boldsymbol{x}_t}{1-t}.
\tag{14}
$$

Substituting equation 14 into equation 13, we obtain

$$
\big\|\boldsymbol{v}_t(\boldsymbol{x}_t \mid X_0, X_1) \;-\; \boldsymbol{v}_t(\boldsymbol{x}_t)\big\|^2 \;=\; \frac{\big\|X_1 - \mathbb{E}[\,X_1 \mid X_t = \boldsymbol{x}_t\,]\big\|^2}{(1-t)^2}.
$$

Assuming that

$$
X_1^{\mathrm{pred}}(\boldsymbol{x}_t, t) \;=\; \mathbb{E}[\,X_1 \mid X_t = \boldsymbol{x}_t\,] \quad \text{and} \quad \boldsymbol{v}_t^{\mathrm{pred}}(\boldsymbol{x}_t) \;=\; \boldsymbol{v}_t(\boldsymbol{x}_t).
$$

Consequently,

$$
\big\|X_1 - X_1^{\mathrm{pred}}(\boldsymbol{x}_t, t)\big\|^2 \;=\; (1-t)^2 \big\|\boldsymbol{v}_t(\boldsymbol{x}_t \mid X_0, X_1) \;-\; \boldsymbol{v}_t^{\mathrm{pred}}(\boldsymbol{x}_t)\big\|^2.
$$

$\qquad\square$

## B.2 Reward as Classifier Guidance

We begin by citing a lemma from the Guided Flows paper [88].

**Lemma B.2.** *Let $p_t(\boldsymbol{x}|\boldsymbol{y})$ be a Gaussian Path defined by a scheduler $(\alpha_t, \sigma_t)$, i.e., $p_t(\boldsymbol{x}|x_0) = \mathcal{N}(\boldsymbol{x}|\alpha_t x_0, \sigma_t^2 I)$ where $\boldsymbol{y} \in \mathbb{R}^k$ is a conditioning variable, then its generating velocity field $\boldsymbol{v}_t(\boldsymbol{x}|\boldsymbol{y})$ is related to the score function $\nabla \log p_t(\boldsymbol{x}|\boldsymbol{y})$ by*

$$
\boldsymbol{v}_t(\boldsymbol{x}|\boldsymbol{y}) = a_t \boldsymbol{x} + b_t \nabla \log p_t(\boldsymbol{x}|\boldsymbol{y}),
\tag{15}
$$

*where*

$$
a_t = \frac{\dot{\alpha}_t}{\alpha_t}, \qquad b_t = (\dot{\alpha}_t \sigma_t - \alpha_t \dot{\sigma}_t)\frac{\sigma_t}{\alpha_t}.
\tag{16}
$$

Seed Appendix A.61 of the Guided Flows paper [88] for detailed derivations.

If we define

$$
\begin{aligned}
\tilde{v}_t(\boldsymbol{x}_t|\boldsymbol{y}) &= \boldsymbol{v}_t(\boldsymbol{x}_t|\boldsymbol{y}) + w[a_t\boldsymbol{x}_t + b_t\nabla r(\boldsymbol{x}_t, \boldsymbol{y}) - \boldsymbol{v}_t(\boldsymbol{x}_t|\boldsymbol{y})] \\
&= \boldsymbol{v}_t(\boldsymbol{x}_t|\boldsymbol{y}) + w[a_t\boldsymbol{x}_t + b_t\nabla \log \exp(r(\boldsymbol{x}_t, \boldsymbol{y})) - \boldsymbol{v}_t(\boldsymbol{x}_t|\boldsymbol{y})] \\
&= (1-w)\boldsymbol{v}_t(\boldsymbol{x}_t|\boldsymbol{y}) + w[a_t\boldsymbol{x}_t + b_t\nabla \log \exp(r(\boldsymbol{x}_t, \boldsymbol{y}))] \\
&= a_t\boldsymbol{x}_t + b_t\nabla\big[(1-w)\log p_t(\boldsymbol{x}_t|\boldsymbol{y}) + w\log \exp(r(\boldsymbol{x}_t, \boldsymbol{y}))\big] \\
&= a_t\boldsymbol{x}_t + b_t\nabla \log \tilde{p}_t(\boldsymbol{x}_t|\boldsymbol{y})
\end{aligned}
\tag{17}
$$

where $\tilde{p}_t(\boldsymbol{x}_t|\boldsymbol{y}) \propto p_t(\boldsymbol{x}_t|\boldsymbol{y})^{1-w}[\exp(r(\boldsymbol{x}_t, \boldsymbol{y}))]^w$. We change our goal from sampling from the distribution $p_t(\boldsymbol{x}_t|\boldsymbol{y})$ to sampling from the distribution $\tilde{p}_t(\boldsymbol{x}_t|\boldsymbol{y})$.

We note that this analysis shows that Reward Guided Flows are guaranteed to sample from $\tilde{q}(\cdot|\boldsymbol{y})$ at time $t = 1$ if the probability path $\tilde{p}_t(\cdot|\boldsymbol{y})$ is close to the marginal probability path $\int p_t(\cdot|\boldsymbol{x}_1)\tilde{q}(\boldsymbol{x}_1|\boldsymbol{y})dx_1$, but it is not clear to what extent this assumption holds in practice. This also mens that $\tilde{p}_t(\boldsymbol{x}_t|\boldsymbol{y})$ is also a marginal gaussian path defined by $p_t(\boldsymbol{x}|\boldsymbol{x}_1) = \mathcal{N}(\boldsymbol{x}|\alpha_t\boldsymbol{x}_1, \sigma_t^2 I)$.

Simlirly, if we define

$$
\begin{aligned}
\tilde{v}_t(\boldsymbol{x}_t|\boldsymbol{y}) &= \boldsymbol{v}_t(\boldsymbol{x}_t|\boldsymbol{y}) + wb_t \nabla r(\boldsymbol{x}_t, \boldsymbol{y}) \\
&= \boldsymbol{v}_t(\boldsymbol{x}_t|\boldsymbol{y}) + wb_t \nabla \log \exp(r(\boldsymbol{x}_t, \boldsymbol{y})) \\
&= a_t \boldsymbol{x}_t + b_t \nabla \left[ \log p_t(\boldsymbol{x}_t|\boldsymbol{y}) + w \log \exp(r(\boldsymbol{x}_t, \boldsymbol{y})) \right] \\
&= a_t \boldsymbol{x}_t + b_t \nabla \log \tilde{p}_t(\boldsymbol{x}_t|\boldsymbol{y})
\end{aligned}
\tag{18}
$$

where $\tilde{p}_t(\boldsymbol{x}_t|\boldsymbol{y}) \propto p_t(\boldsymbol{x}_t|\boldsymbol{y})[\exp(r(\boldsymbol{x}_t, \boldsymbol{y}))]^w$. We change our goal from sampling from the distribution $p_t(\boldsymbol{x}_t|\boldsymbol{y})$ to sampling from the distribution $\tilde{p}_t(\boldsymbol{x}_t|\boldsymbol{y})$.

**Reward Guidance for Rectified Flow.** Rectified Flow [48] is also a Gaussian path defined by

$$
\boldsymbol{x}_t = (1-t)\boldsymbol{x}_0 + t\boldsymbol{x}_1
\tag{19}
$$

where $\boldsymbol{x}_1$ is from normal Gaussian distribution. Then

$$
p_t(\boldsymbol{x} \mid \boldsymbol{x}_0) = \mathcal{N}(\boldsymbol{x}|(1-t)\boldsymbol{x}_0, t^2 I)
$$

where $\alpha_t = 1 - t, \sigma_t = t$. Then we get

$$
a_t = \frac{1}{t-1}, b_t = \frac{t}{t-1}.
\tag{20}
$$

Eq. 17 becomes

$$
\tilde{v}_t(\boldsymbol{x}_t|\boldsymbol{y}) = \boldsymbol{v}_t(\boldsymbol{x}_t|\boldsymbol{y}) + w[\frac{1}{1-t}\boldsymbol{x}_t + \frac{t}{1-t}\nabla r(\boldsymbol{x}_t, \boldsymbol{y}) + \boldsymbol{v}_t(\boldsymbol{x}_t|\boldsymbol{y})].
\tag{21}
$$

Eq. 18 becomes

$$
\tilde{v}_t(\boldsymbol{x}_t|\boldsymbol{y}) = \boldsymbol{v}_t(\boldsymbol{x}_t|\boldsymbol{y}) - w\frac{t}{1-t}\nabla r(\boldsymbol{x}_t, \boldsymbol{y}).
\tag{22}
$$

We use Eq. 17 or Eq. 18 to guide inference through reward model.

## C Pseudo-code of Flow-DPO and Flow-NRG

The Pytorch-style implementation of the Flow-DPO loss (Eq. 6) is shown below:

```python
def loss(model, ref_model, x_w, x_l, c, beta):
    """
    # model: Flow model that takes prompt condition c and timestep as inputs and
        predicts velocity
    # ref_model: Frozen initialization of the model
    # x_w: Preferred Image (latents in this work)
    # x_l: Non-Preferred Image (latents in this work)
    # c: Conditioning (text in this work)
    # beta: Regularization Parameter
    """
    timestep = torch.rand(len(x_w))
    noise = torch.randn_like(x_w)
    noisy_x_w = (1 - timestep) * x_w + timestep * noise
    noisy_x_l = (1 - timestep) * x_l + timestep * noise
    velocity_w_pred = model(noisy_x_w, c, timestep)
    velocity_l_pred = model(noisy_x_l, c, timestep)
    velocity_ref_w_pred = ref_model(noisy_x_w, c, timestep)
    velocity_ref_l_pred = ref_model(noisy_x_l, c, timestep)
    velocity_w = noise - x_w
    velocity_l = noise - x_l

    model_w_err = (velocity_w_pred - velocity_w).norm().pow(2)
    model_l_err = (velocity_l_pred - velocity_l).norm().pow(2)
    ref_w_err = (velocity_ref_w_pred - velocity_w).norm().pow(2)
    ref_l_err = (velocity_ref_l_pred - velocity_l).norm().pow(2)
    w_diff = model_w_err - ref_w_err
    l_diff = model_l_err - ref_l_err
    inside_term = -0.5 * beta * (w_diff - l_diff)
    loss = -1 * log(sigmoid(inside_term))
    return loss
```

The Pytorch-style implementation of the reward guidance (Eq. 11) is shown below:

```python
def reward_guidance(model, reward_model, prompt_embeds, latents, timesteps,
    reward_weight, rg_scale, cfg_scale):
    """
    # model: Flow model that predicts velocity given latents and conditions
    # reward_model: Model that evaluates the quality of latents based on prompt
        embeddings and timestep
    # prompt_embeds: Embeddings of the text prompts
    # latents: Initial noise
    # timesteps: Sequence of timesteps
    # reward_weight: weighting coefficient of multi-dimensional rewards
    # rg_scale: scale factor for reward guidance
    # cfg_scale: scale factor for classifier free guidance
    """
    dts = timesteps[:-1] - timesteps[1:]
    for i, t in enumerate(timesteps):
        v_pred = model(latents, prompt_embeds, t)
        if cfg_scale != 1.0:
            v_pred_uncond = model(latents, None, t) # unconditional prediction
            v_pred = v_pred_uncond + cfg_scale * (v_pred - v_pred_uncond)
        latents = latents.detach().requires_grad_(True)
        reward = reward_model(latents, prompt_embeds, t)
        reward = (reward * reward_weight).sum()
        reward_guidance = torch.autograd.grad(reward, latents)
        if t != 1:
            v_pred = v_pred - rg_scale * t / (1 - t) * reward_guidance
        latents = latents - dts[i] * v_pred
    return latents
```

# D  Architecture of Internal Video Diffusion Model

Our work employs an internal text-to-video foundation model, which is a Transformer-based latent diffusion model [53] with 1B parameters. The model integrates a 3D VAE to encode video data into a latent space, alongside a Transformer-based video diffusion model. Each Transformer block is instantiated as a sequence of 2D spatial-attention, 3D self-attention, cross-attention, and FFN modules. Importantly, the model is trained under Flow Matching framework. An illustration of the model architecture is provided in Fig 7.

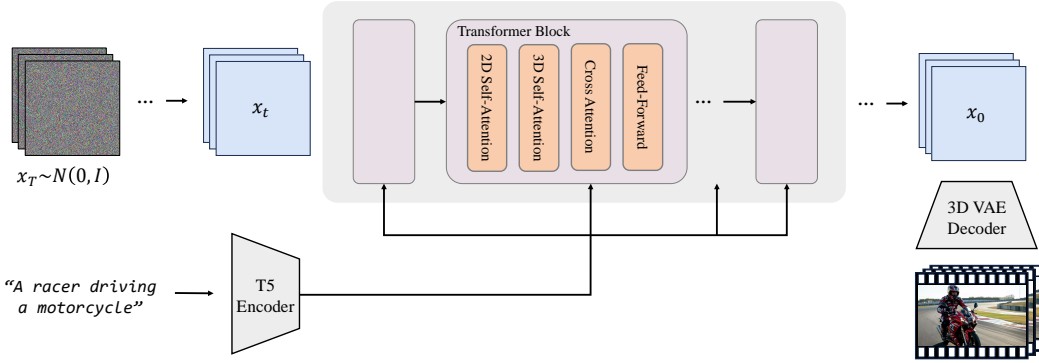

Figure 7: Architecture of Our Internal Video Latent Diffusion Model Backbone.

# E  Ablation Study of Our Reward Model

**Ablation of VideoReward**   To further validate the key design choices of our reward model discussed in Sec.3.2, we conduct an ablation study on the evaluation benchmark, providing a quantitative supplement to our analysis in Tab. 6. We compare three reward model variants: *regression-style*, *Bradley-Terry*, and *Bradley-Terry-With-Tied*. The BT model slightly outperforms the regression-style model, likely due to the advantages of pairwise annotations, which better capture ranking relationships and are more robust to annotation noise. The BTT model matches the BT model on ties-excluded accuracy but significantly improves ties-included accuracy, as its explicit handling of tied pairs helps it learn a more robust decision boundary, capturing neutral relationships in ambiguous cases. Additionally, we find that using separate tokens for each reward attribute, instead of a shared last token further improves performance. This design better represents distinct reward aspects, enhancing alignment with human preferences.

Table 6: Ablation study on reward model type and seprate tokens. **Bold**: Best Performance.

| Variants | RM Type | Separate Tokens | GenAI-Bench Overall Accuracy w/ Ties | w/o Ties | VideoGen-RewardBench Overall Accuracy w/ Ties | w/o Ties | VQ Accuracy w/ Ties | w/o Ties | MQ Accuracy w/ Ties | w/o Ties | TA Accuracy w/ Ties | w/o Ties |
|---|---|---|---|---|---|---|---|---|---|---|---|---|
| I | Regression | | 48.28 | 71.13 | 58.39 | 70.16 | 54.23 | 73.61 | 61.16 | 65.56 | 52.60 | 70.95 |
| II | BT | | 47.74 | 71.21 | 61.22 | 72.58 | 52.33 | **77.10** | 59.43 | 73.50 | 53.06 | 71.62 |
| III | BTT | | 48.27 | 70.89 | **61.50** | 73.39 | **60.52** | 76.31 | 64.64 | 72.40 | 53.55 | 72.12 |
| IV | BTT | ✓ | **49.41** | **72.89** | 61.26 | **73.59** | 59.68 | 75.66 | **66.03** | **74.70** | **53.80** | **72.20** |

**Ablation on Noisy-Latent Reward Training.**   We apply reward guidance using only MQ rewards on TA-Hard prompts. When trained with noised latents, the reward model successfully guides generation, yielding an MQ win rate of 74.6. By contrast, the model trained on clean latents fails to offer meaningful guidance for intermediate noised latents, achieving only a 38.6 win rate and underperforming even the unguided baseline. Following Yu et al. [81], we also try directly predicting $x_0$ from $x_t$ and then backpropagating via $r(x_0, y)$. However, this method produces unusable videos without meaningful content.

# F  Statistics of Training Data

In Tab. 1 and Fig. 8, we provide a comprehensive statistics of our 182k-sized training dataset.

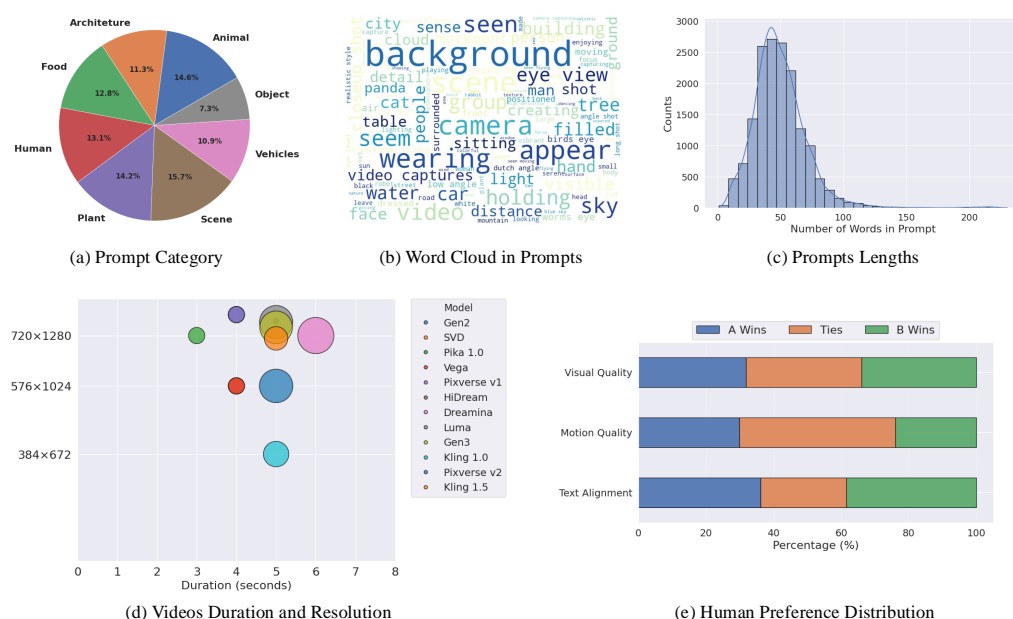

(a) Prompt Category      (b) Word Cloud in Prompts      (c) Prompts Lengths

(d) Videos Duration and Resolution      (e) Human Preference Distribution

Figure 8: Statistics of our training data.

Previous reward models (e.g., VideoScore [19], LIFT [73]) were trained on datasets with mostly low resolutions (≤480p) and short durations (2s), whereas ours include recent T2V models with higher resolutions (≥720p) and longer clips (3–6s). The dataset statistics are shown in Tab. 7.

Table 7: Comparison of video preference datasets in terms of spatial resolution and temporal duration.

| Dataset | Resolution | Duration |
|---|---|---|
| *Existing Preference Datasets* | | |
| VideoFeedback [19] | $256 \times 256 - 576 \times 1024$ | $1s - 3s$ |
| VideoDPO [47] | $320 \times 512$ | $2s$ |
| LiFT-HRA [73] | $480 \times 720$ | $6s$ |
| *Our Datasets* | | |
| Ours | $384 \times 672 - 768 \times 1408$ | $3s - 6s$ |

# G  Details of Human Annotation

We provide additional details regarding the annotation process. First, annotators are provided with detailed scoring guidelines and undergo training sessions to ensure they fully understand the criteria; Tab 8 summarizes the key points for each dimension as outlined in the guidelines. Reference examples are provided to help annotators better grasp the evaluation standards. Each sample is evaluated by three independent annotators. For training and validation sets, annotators provide pairwise preference annotations and pointwise scores for Visual Quality (VQ), Motion Quality (MQ), and Tempotal Alignment (TA). For VideoGen-RewardBench, annotators evaluate the same three aspects along with an additional Overall Quality, using only pairwise preferences. In cases where the annotators disagree on a sample, an additional reviewer is tasked with resolving the discrepancy. The final label is determined on the basis of the reviewer's evaluation, ensuring consistency across the dataset. Furthermore, during the annotation process, all annotators are instructed to flag any content deemed unsafe. Videos identified as unsafe are excluded from the dataset, ensuring the safety of the data used for training and evaluation.

Table 8: Key points summary outlined in annotation guidelines for each evaluation dimension.

| Evaluation Dimension | Key Points Summary |
|---|---|
| Visual Quality | Considering the following dimensions introduced by **non-dynamic** factors:
- **Image Reasonableness**: The image should be objectively reasonable.
- **Clarity**: The image should be clear and visually sharp.
- **Detail Richness**: The level of intricacy in the generation of details.
- **Aesthetic Creativity**: The generated videos should be aesthetically pleasing.
- **Safety**: The generated video should not contain any disturbing or uncomfortable content. |
| Motion Quality | Considering the following dimensions in the **dynamic** process of the video:
- **Dynamic Stability**: The continuity and stability between frames.
- **Dynamic Reasonableness**: The dynamic movement should align with natural physical laws.
- **Motion Aesthetic Quality**: The dynamic elements should be harmonious and not stiff.
- **Naturalness of Dynamic Fusion**: The edges should be clear during the dynamic process.
- **Motion Clarity**: The motion should be easy to identify.
- **Dynamic Degree**: The movement should be clear, avoiding still scenes. |
| Text Alignment | Considering the **relevance** to the input text prompt description.
- **Subject Relevance** Relevance to the described subject characteristics and subject details.
- **Dynamic Information Relevance**: Relevance to actions and postures as described in the text.
- **Environmental Relevance**: Relevance of the environment to the input text.
- **Style Relevance**: Relevance to the style descriptions, if exists.
- **Camera Movement Relevance**: Relevance to the camera descriptions, if exists. |

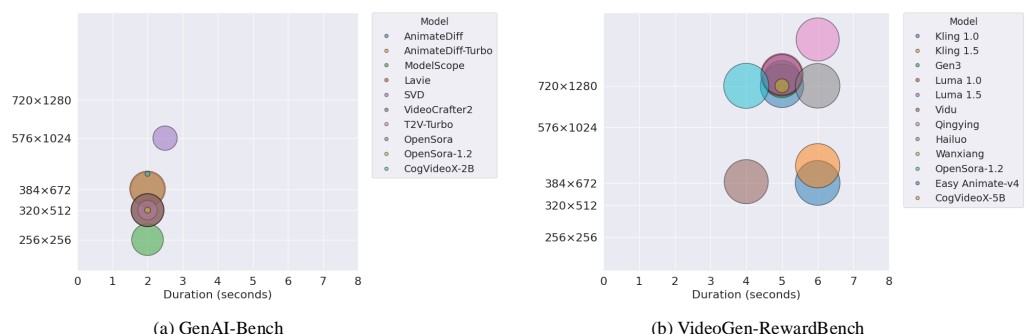

(a) GenAI-Bench      (b) VideoGen-RewardBench

Figure 9: Video Duration and Resolution in GenAI-Bench and VideoGen-Reward Bench

# H  Details of Reward Model Evaluation

## H.1  Evaluation Benchmarks

We evaluate our reward model using two benchmarks: GenAI-Bench [29] and VideoGen-RewardBench. **GenAI-Bench** is employed to assess the accuracy of the reward model in **evaluating pre-SOTA-era T2V models**, while **VideoGen-RewardBench** is used to evaluate its performance on **modern T2V models**. In this subsection, we describe both benchmarks, highlighting key parameters and differences in Fig 9 and Tab. 9. We also visualize the model coverage across the training sets of different baselines and the two evaluation benchmarks, as shown in the Fig 10.

**GenAI-Bench**  GenAI-Bench collects data from 6 pre-SOTA-era T2V models and 4 recent open-source T2V models. Human annotations for overall quality are obtained through GenAI-Arena, resulting in a benchmark consisting of 10 T2V models, 508 prompts, and 1.9k pairs. As the videos

Table 9: Comparison between GenAI-Bench and VideoGen-RewardBench. Eariler Models indicates that pre-Sora-era T2V models, and Modern Models indicates that T2V models after Sora.

| Benchmark | Prompts and Sampled Videos | | | | | Human Preference Annotations | |
|---|---|---|---|---|---|---|---|
| | #Samples | #Prompts | #Earlier Models | #Modern Models | #Duration | #Annotations | #Dimensions |
| GenAI-Bench | 3784 | 508 | 7 (Open-Source) | 3 (Open-Source) | 2s - 2.5s | 1891 | 1 |
| VideoGen-RewardBench | 4923 | 420 | 0 | 3 (Open-Source) 9 (Close-Source) | 4s - 6s | 26457 | 4 |

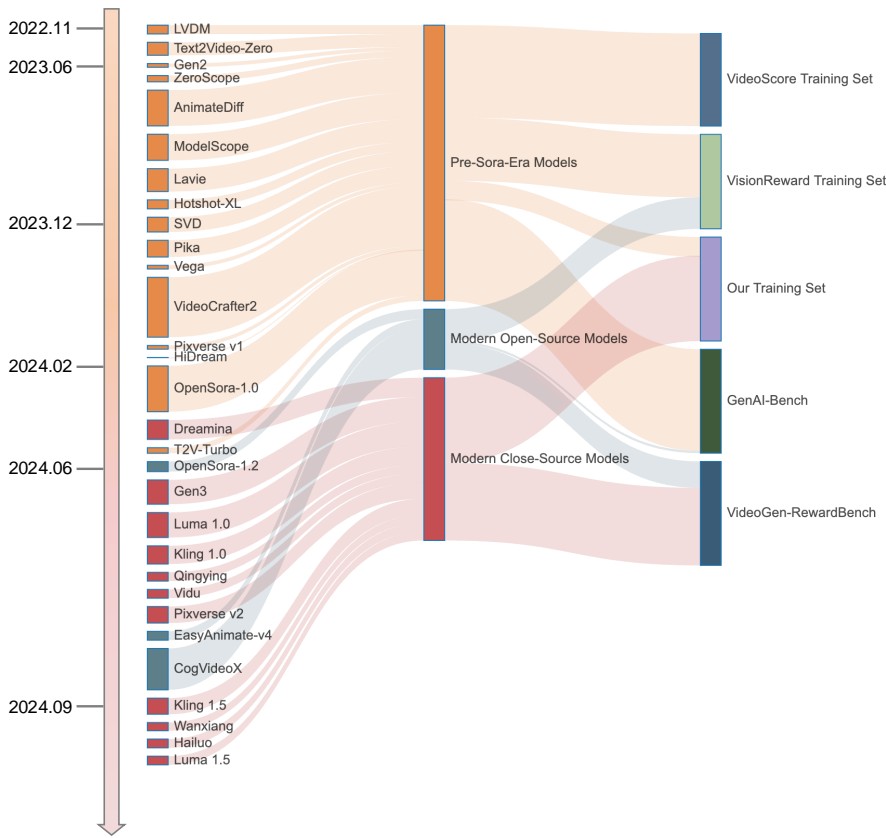

Figure 10: The model coverage across the training sets of different baselines and the two evaluation benchmarks. VideoScore, VisionReward, and GenAI-Bench primarily focus on pre-SoRA-era models, while our training set and VideoGen-RewardBench concentrate on state-of-the-art T2V models.

in GenAI-Bench predominantly originate from earlier video generation models, they typically have lower resolutions (most around 320x512) and shorter durations (2s-2.5s). We consider GenAI-Bench as a benchmark to assess the performance of reward models on early-generation T2V models.

**VideoGen-RewardBench**    VideoGen-Eval [84] has open-sourced a dataset containing videos generated by 9 closed-source and 3 open-source models, designed to qualitatively visualize performance differences across models. Due to its high-quality data, broad coverage of the latest advancements in T2V models, and third-party nature, we leverage VideoGen-Eval to create a fair benchmark, VideoGen-RewardBench, for evaluating reward models' performance on modern T2V models. We manually construct 26.5k video pairs and hire annotators to assess each pair's Visual Quality, Motion Quality, Text Alignment, and Overall Quality, providing preference labels. Ultimately, VideoGen-RewardBench includes 12 T2V models, 420 prompts, and 26.5k pairs. This benchmark represents human preferences for state-of-the-art models, with videos featuring higher resolutions (480x720 - 576x1024), longer durations (4s - 6s), and improved quality. We use VideoGen-RewardBench as the primary benchmark to evaluate reward models' performance on modern T2V models.

## H.2 Comparison Methods

**Random**    To eliminate the influence of metric calculations and benchmark distributions on our evaluation results, we introduce a special baseline: random scores. Specifically, for each triplet *(prompt, video A, video B)*, we randomly sample $r_A$ and $r_B$ from a standard normal distribution, denoted as $r_A, r_B \sim \mathcal{N}(0, 1)$. We then calculate accuracy in the same manner as for the other models. The mathematical expectation of random scores for ties-excluded accuracy is $\mathbb{E}(acc) = \frac{1}{2}$, and the mathematical expectation of ties-included accuracy is $\mathbb{E}(acc) = \max(\frac{1}{3}, p(c = \text{"Ties"}))$.

**VideoScore**    VideoScore [19] adopts Mantis-Idefics2-8B [28] as its base model and trains with point-wise data using MSE loss to model human preference scores. Since VideoScore predicts scores across multiple dimensions, and its dimension definitions differ from those in VideoGen-RewardBench, we compute both the overall accuracy and the dimension-specific (VQ, MQ, TA) accuracy by averaging the scores of five dimensions when conduct evaluation GenAI-Bench and VideoGen-RewardBench, consistent with the evaluation strategy outlined in their paper. The training data for VideoScore predominantly comes from pre-SOTA-era models, which explains its relatively better performance on GenAI-Bench, while accounts for the significant performance drop on VideoGen-RewardBench.

**LiFT**    LiFT [73] adopts VILA-1.5-40B [43] as its base model and employs a VLM-as-a-judge approach. The reward model is trained through instruction tuning with inputs, preference scores along with a critic. The model generates video scores and reasons through next-token prediction. LiFT evaluates videos across three dimensions: Video Fidelity, Motion Smoothness, and Semantic Consistency, which are similar to the dimensions defined in VideoGen-RewardBench. We calculate the overall accuracy using the average scores of these three dimensions and compute the dimension-specific accuracy using the corresponding dimensional scores. LiFT predicts discrete scores on a 1-3 scale, which often leads to ties in pairwise comparisons. When calculating accuracy without ties, we randomly convert the predicted tie labels to chosen/rejected with a 50% probability, indicating that the model is unable to distinguish the relative quality between the two samples.

**VisionReward**    VisionReward [76] adopts CogVLM2-Video-12B[23] as the base model and is trained to answer a set of judgment questions about the video with a binary "yes" or "no" response using cross-entropy loss. During inference, VisionReward evaluates 64 checklist items, providing converted into 1/0 scores. The final score is computed as the weighted average of these individual responses. We use the final score to calculate both the overall accuracy and the VQ/MQ/TA accuracy. VisionReward's training data includes models from the pre-SOTA era models [8] as well as recent open-source T2V models [89, 79]. It performs well on GenAI-Bench and demonstrates reasonable capabilities on VideoGen-RewardBench.

**Our Reward Model**    We adopts QWen2-VL-2B [72] as the base model and train it with pair-wise data using BTT loss in Eq. 2. Scores are normalized on the validation set and averaged to obtain overall scores for evaluation and optimization. When evaluating on VideoGen-RewardBench, we sample videos at 2 FPS and a resolution of 448×448, consistent with the training settings. We **calculate the overall accuracy by averaging the scores** across the three dimensions, and compute dimension-specific accuracies using the respective scores. For GenAI-Bench, we sample videos at 2 FPS and a resolution of 256×256, as the minimum resolution in GenAI-Bench is 256×256. Given the significant disparities in visual quality and motion between the GenAI-Bench videos and our training data, **we utilize only the predicted TA scores to calculate the overall score**.

## H.3 Evaluation Metrics

Similarly to VisionReward [76], we report two accuracy metrics: ties-included accuracy [11] and ties-excluded accuracy. For ties-excluded accuracy, we exclude all data labeled as *"ties"* and use only data labeled as *"A wins"* or *"B wins"* for calculation. Since all competitors predict scores based on pointwise samples, we compute the rewards for each pair, convert the relative reward relationships into binary labels, and calculate classification accuracy. For ties-included accuracy, we adopt the tie calibration algorithm proposed in Algorithm 1 by Deutsch et al. [11]. This method traverses all possible tie thresholds, calculates three-class accuracy for each threshold, and selects the highest accuracy as the final metric.

# I  Hyperparameters

In all alignment experiments, we applied LoRA to fine-tune the transformer models' linear layers, as our findings indicate that full parameter fine-tuning can degrade the model's performance or potentially lead to model collapse.

Table 10: Hyperparameters for alignment algorithms

| Algorithm-agnostic hyperparameters for SFT, Flow-RWR, Flow-DPO | |
|---|---|
| Training strategy | LoRA [24] |
| LoRA alpha | 128 |
| LoRA dropout | 0.0 |
| LoRA R | 64 |
| LoRA target-modules | q_proj,k_proj,v_proj,o_proj |
| Optimizer | Adam [30] |
| Learning rate | 5e-6 |
| Epochs | 1 |
| Batch size | 64 |
| GPUs | 16 NVIDIA A800@80G |
| **Flow-DPO** | |
| $\beta$ | 500 |

Table 11: Hyperparameters for reward modeling.

| VLM | |
|---|---|
| Training strategy | Full training for vision encoder LoRA for language model |
| LoRA alpha | 128 |
| LoRA dropout | 0.0 |
| LoRA R | 64 |
| LoRA target-modules | Linear layers in language model |
| Optimizer | Adam [30] |
| Learning rate | 2e-6 |
| Epochs | 2 |
| Batch size | 32 |
| GPUs | 8 NVIDIA A800@80G |
| $\theta$ in Eq. 2 | 5.0 |
| **VDM** | |
| Training strategy | Full training |
| Optimizer | Adam [30] |
| Learning rate | 5e-6 |
| Epochs | 2 |
| Batch size | 144 |
| Reward Dimension | 3 |
| GPUs | 8 NVIDIA A800@80G |

# J  Additional Qualitative Results

We present additional qualitative results generated by both the original model and the Flow-DPO aligned model, as shown in Fig.11 and Fig.12.

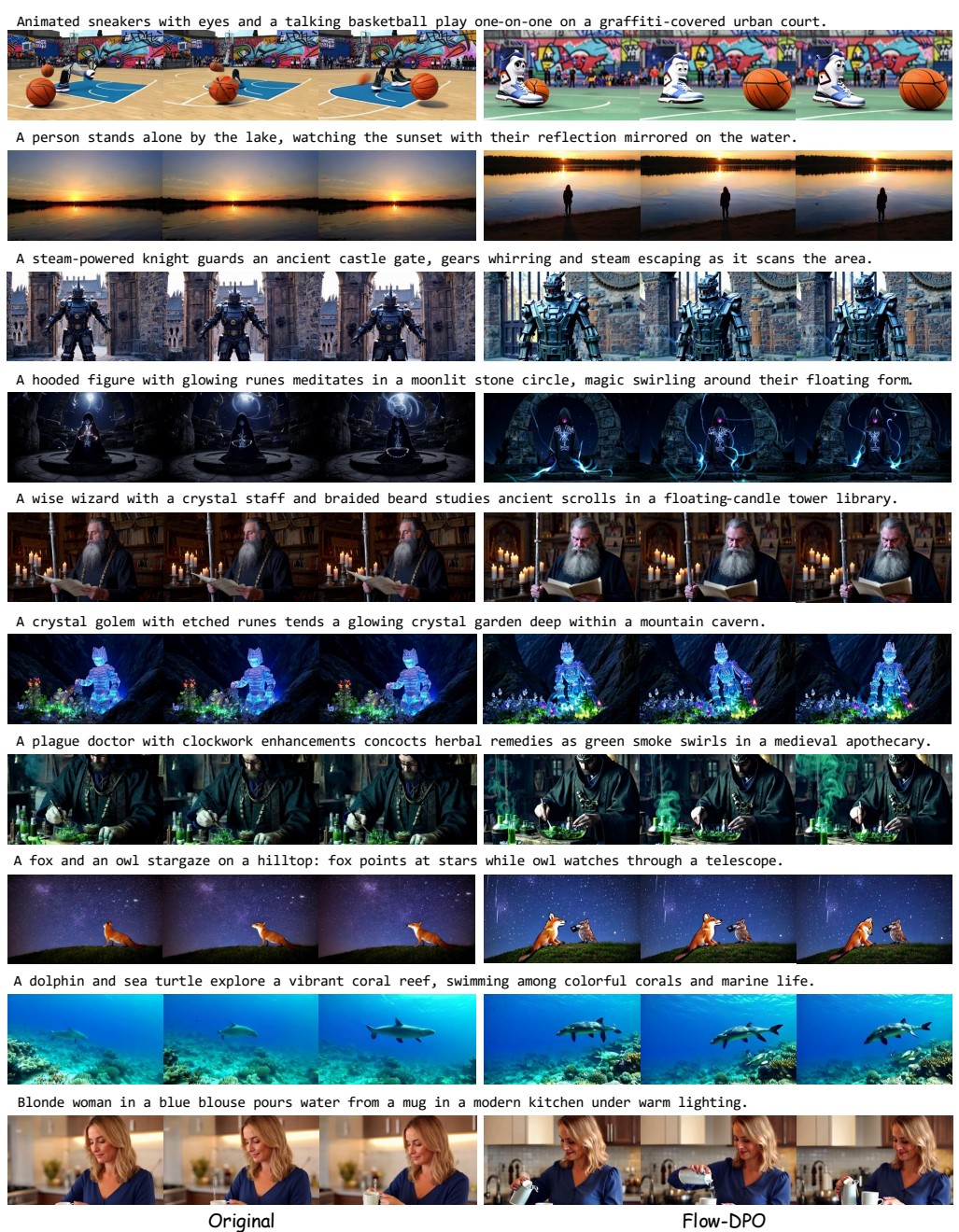

Figure 11: Additional visual comparison of videos generated by the original model and the Flow-DPO aligned model.

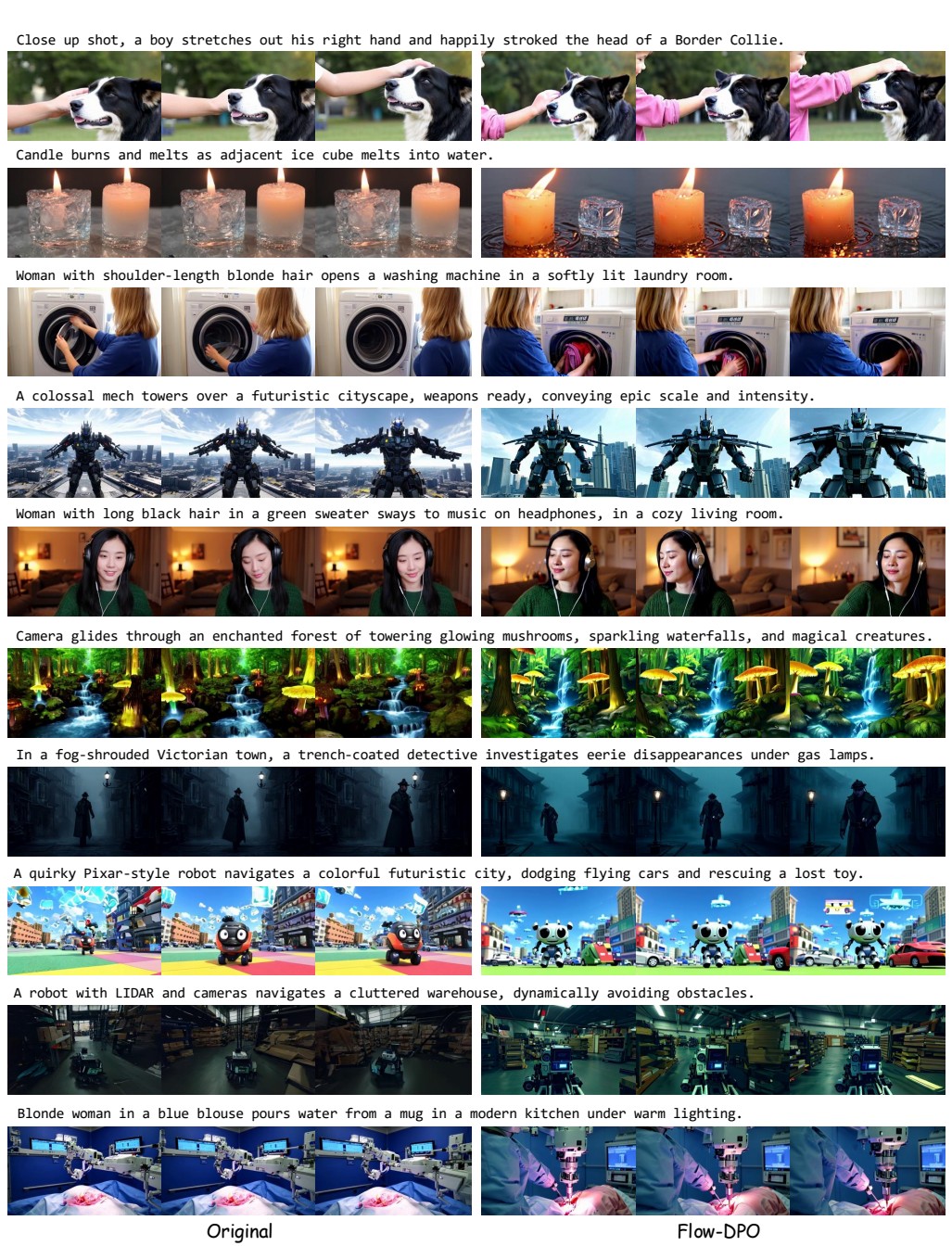

Figure 12: Additional visual comparison of videos generated by the original model and the Flow-DPO aligned model.

# K Input Template for Reward Model

> **Full Input Template**
>
> `[VIDEO]` You are tasked with evaluating a generated video based on three distinct criteria: Visual Quality, Motion Quality, and Text Alignment. Please provide a rating from 0 to 10 for each of the three categories, with 0 being the worst and 10 being the best. Each evaluation should be independent of the others.
>
> **Visual Quality:**
> Evaluate the overall visual quality of the video, with a focus on static factors. The following sub-dimensions should be considered:
> - **Reasonableness:** The video should not contain any significant biological or logical errors, such as abnormal body structures or nonsensical environmental setups.
> - **Clarity:** Evaluate the sharpness and visibility of the video. The image should be clear and easy to interpret, with no blurring or indistinct areas.
> - **Detail Richness:** Consider the level of detail in textures, materials, lighting, and other visual elements (e.g., hair, clothing, shadows).
> - **Aesthetic and Creativity:** Assess the artistic aspects of the video, including the color scheme, composition, atmosphere, depth of field, and the overall creative appeal. The scene should convey a sense of harmony and balance.
> - **Safety:** The video should not contain harmful or inappropriate content, such as political, violent, or adult material. If such content is present, the image quality and satisfaction score should be the lowest possible.
>
> Please provide the ratings of Visual Quality: `<|VQ_reward|>`
> END
>
> **Motion Quality:**
> Assess the dynamic aspects of the video, with a focus on dynamic factors. Consider the following sub-dimensions:
> - **Stability:** Evaluate the continuity and stability between frames. There should be no sudden, unnatural jumps, and the video should maintain stable attributes (e.g., no fluctuating colors, textures, or missing body parts).
> - **Naturalness:** The movement should align with physical laws and be realistic. For example, clothing should flow naturally with motion, and facial expressions should change appropriately (e.g., blinking, mouth movements).
> - **Aesthetic Quality:** The movement should be smooth and fluid. The transitions between different motions or camera angles should be seamless, and the overall dynamic feel should be visually pleasing.
> - **Fusion:** Ensure that elements in motion (e.g., edges of the subject, hair, clothing) blend naturally with the background, without obvious artifacts or the feeling of cut-and-paste effects.
> - **Clarity of Motion:** The video should be clear and smooth in motion. Pay attention to any areas where the video might have blurry or unsteady sections that hinder visual continuity.
> - **Amplitude:** If the video is largely static or has little movement, assign a low score for motion quality.
>
> Please provide the ratings of Motion Quality: `<|MQ_reward|>`
> END
>
> **Text Alignment:**
> Assess how well the video matches the textual prompt across the following sub-dimensions:
> - **Subject Relevance** Evaluate how accurately the subject(s) in the video (e.g., person, animal, object) align with the textual description. The subject should match the description in terms of number, appearance, and behavior.
> - **Motion Relevance:** Evaluate if the dynamic actions (e.g., gestures, posture, facial expressions like talking or blinking) align with the described prompt. The motion should match the prompt in terms of type, scale, and direction.
> - **Environment Relevance:** Assess whether the background and scene fit the prompt. This includes checking if real-world locations or scenes are accurately represented, though some stylistic adaptation is acceptable.
> - **Style Relevance:** If the prompt specifies a particular artistic or stylistic style, evaluate how well the video adheres to this style.
> - **Camera Movement Relevance:** Check if the camera movements (e.g., following the subject, focus shifts) are consistent with the expected behavior from the prompt.
>
> Textual prompt - `[PROMPT]`
> Please provide the ratings of Text Alignment: `<|TA_reward|>`
> END

# L   Prompt Subset of TA-Hard

A rabbit and a turtle racing on a track. The rabbit is sprinting ahead, while the turtle is steadily moving along. Spectators are cheering from the sidelines, and a finish line is visible in the distance.

A lion and a zebra playing soccer on a grassy field. The lion is dribbling the ball, while the zebra is trying to block it. The field is surrounded by trees, and other animals are watching the game.

A fox and an owl stargazing together on a hilltop. The fox is lying on its back, pointing at the stars, while the owl is perched on a nearby branch, looking through a telescope. The night sky is clear, with countless stars twinkling.

A dolphin and a sea turtle exploring a coral reef. The dolphin is swimming gracefully, while the sea turtle is gliding slowly beside it. The coral reef is vibrant with colorful corals and various marine life.

A dolphin and a whale singing together in the ocean. The dolphin is leaping out of the water, while the whale is producing deep, melodic sounds. The ocean is vast and blue, with the sun setting on the horizon.

A fox and a rabbit playing a duet on a piano in a forest clearing. The fox is playing the melody, while the rabbit is accompanying with harmony. The forest is alive with the sounds of nature, and other animals are gathered to listen.

A squirrel and a chipmunk building a treehouse in a large oak tree. The squirrel is hammering nails, while the chipmunk is holding a blueprint. The tree is tall and sturdy, with branches full of leaves.

A robot with glowing blue eyes and a human with a cybernetic arm playing basketball in a futuristic gym. The robot is dribbling the ball with precision, while the human is preparing to block the shot. The gym is equipped with advanced technology and holographic scoreboards.

A knight in shining armor and a wizard with a long, flowing beard practicing archery in a medieval courtyard. The knight is aiming at a target with a longbow, while the wizard is using magic to guide the arrows. The courtyard is surrounded by stone walls and blooming flowers.

A talking apple with eyes and a mouth, and a singing banana with legs hosting a talent show in a vibrant theater. The apple is the judge, giving feedback to contestants, while the banana is the host, entertaining the audience with jokes and songs. The theater is filled with colorful lights and excited spectators.

A pirate with a wooden leg and a mermaid with a shimmering tail playing a duet on a grand piano in an underwater cave. The pirate is playing the melody, while the mermaid is accompanying with harmony. The cave is illuminated by bioluminescent sea creatures, creating a magical atmosphere.

A superhero with a cape and a detective with a magnifying glass solving a mystery in a bustling city. The superhero is flying above the streets, scanning for clues, while the detective is examining evidence on the ground. The city is alive with activity, with skyscrapers towering overhead.

A chef with a tall hat and a robot with multiple arms cooking a gourmet meal in a state-of-the-art kitchen. The chef is chopping vegetables with precision, while the robot is simultaneously stirring, frying, and baking. The kitchen is equipped with the latest culinary technology, creating a seamless cooking experience.

A painter with a beret and a poet with a quill creating art in a sunlit studio. The painter is working on a vibrant canvas, while the poet is writing verses inspired by the artwork. The studio is filled with natural light and creative energy, with art supplies scattered around.

A spider with a square face and a green-furred puppy having a playful fight in a whimsical garden. The spider is using its web to swing around, while the puppy is playfully nipping at the spider's legs. The garden is filled with oversized flowers and colorful mushrooms.

A talking teapot with a mustache and a dancing teacup with legs performing a tea ceremony in an enchanted forest. The teapot is pouring tea, while the teacup is twirling and dancing around. The forest is magical, with glowing plants and twinkling lights.

A robot with a television screen for a head and a toaster with arms and legs having a cooking competition in a retro kitchen. The robot is displaying recipes on its screen, while the toaster is popping out perfectly toasted bread. The kitchen is styled with vintage appliances and checkered floors.

A pair of animated scissors with eyes and a mouth and a roll of tape with tiny arms and legs wrapping presents in a festive workshop. The scissors are cutting wrapping paper with precision, while the tape is sealing the packages with a smile. The workshop is decorated with holiday lights and ornaments.

A pair of animated sneakers with eyes and a mouth and a talking basketball with a face playing a game of one-on-one on an urban basketball court. The sneakers are dribbling and making quick moves, while the basketball is bouncing and trying to score. The court is surrounded by graffiti-covered walls and cheering spectators.

A paper airplane with a scarf and a paper boat with a captain's hat racing in the rain. The airplane glides through the air while the boat sails through puddles.

A basketball with a mohawk and a soccer ball with a bandana playing hopscotch in a playground. The basketball bounces high while the soccer ball rolls smoothly.

A mechanical knight with steam-powered joints standing guard at an ancient castle gate. Gears whir softly as its head turns to scan the surroundings, while steam occasionally escapes from its armor joints.

A wandering alchemist with potion-filled vials clinking on their belt, gathering herbs in an enchanted forest where mushrooms glow and flowers whisper secrets.

A mysterious plague doctor with clockwork enhancements peeking through their dark robes, mixing herbal remedies in a medieval apothecary shop as green smoke swirls from bubbling vials.

## M   Broader Impacts

Our work aims to improve video generation with human feedback. While the primary goal is to improve performance, we recognize that video-generation technologies have broader societal implications. Positive applications may include enhanced creative expression and accessibility tools, thereby benefiting diverse communities. However, the same methods can be misused for creating deceptive or harmful content (e.g., deepfakes), leading to misinformation, reputational harm, and erosion of public trust.

