# OpenReview forum: "Improving Video Generation with Human Feedback"
_NeurIPS.cc/2025/Conference — NeurIPS 2025 poster_

### Official Review · Reviewer_zALS · 2025-06-29

**Clarity:** 3
**Significance:** 3
**Originality:** 1
**Rating:** 4
**Confidence:** 5

**Summary:**

This paper introduce VideoReward, a multi-dimensional video reward model, and examine how annotations and various design choices
impact its rewarding efficacy. It brings three alignment algorithms for flow-based models. These include two training-time strategies: direct preference optimization for flow (Flow-DPO) and reward weighted regression for flow (Flow-RWR), and an inference-time technique, Flow-NRG, which applies reward guidance directly to noisy videos. Experimental results indicate that VideoReward significantly outperforms existing reward models, and Flow-DPO demonstrates superior performance compared to both Flow-RWR and supervised fine-tuning methods.

**Questions:**

Regarding the reward model methodology, do you plan to open-source your dataset and pre-trained reward model? (This won’t affect my evaluation of the paper—it’s merely a suggestion.) For methods like DPO or GRPO, the quality of the constructed dataset often has a more significant impact than the algorithmic details themselves. Open-sourcing the code and data could greatly benefit future research and applications by enabling others to build upon your work.

Currently, training solely on synthetic data introduces the limitations I mentioned earlier for the reward model. However, using real data (e.g., actual video clips) directly often leads to failures in DPO/GRPO training. Have you explored solutions to this issue? Specifically, have you attempted training with real data, and if so, does your method still perform effectively?

**Ethical Concerns:**

["NO or VERY MINOR ethics concerns only"]

**Final Justification:**

Thank the author for the response. After considering the opinions of other reviewers, I have decided to maintain my score.

**Quality:**

3

**Strengths And Weaknesses:**

Strengths:

- Constructed a relatively comprehensive dataset (as listed in Table 1).
- Proposed three effective algorithms specifically designed to enhance video generation quality for flow-matching methods.
- Provided extensive experimental results, effectively demonstrating both the limitations and standout performance of the new reward model.

Weaknesses:

The paper relies solely on model-generated data to construct the training set, which may be problematic. Based on our experience, even state-of-the-art models (e.g., Keling, Veo3, etc.) often fail to produce video clips of fully satisfactory quality, even after multiple sampling attempts. Consequently, we believe this approach makes it difficult for the trained reward model to surpass the performance ceiling of existing models.

---

> ### Author Rebuttal · Authors · 2025-07-31
>
> We sincerely appreciate your insightful comments and efforts in reviewing our manuscript. We respond to each of your comments one by one in what follows.
>
> ---
>
> ` [W1]: Limitations of using model-generated data for reward model training. `
>
>
> We appreciate the reviewer’s observation and agree that current SOTA video generation models still fall short of producing video clips that match the fidelity of real-world videos. However, we would like to clarify the reasoning behind our choice of using model-generated data for training the reward model:
>
> 1. **The goal of the reward model is to capture human preferences among outputs with similar generation quality, rather than to distinguish between real and generated videos.** As the reviewer rightly pointed out, given the current gap between real and generated videos, assessing the quality difference between real and generated videos is often a trivial task. This distinction may not provide meaningful learning signals for the fine-grained preference modeling required in our setting. Constructing such incomparable pairs might hinder the model's ability to learn subtle preference differences.
> 2. **Our reward model is primarily used to guide post-training algorithms such as Flow-DPO or other RL-based methods.** In this context, the reward model needs to capture preference distributions among outputs that are within the capability range of the base generation model, as these are the samples it will be asked to evaluate during alignment. For this reason, we intentionally construct the reward model training set using generations from recent, high-quality T2V models. This ensures the reward model can provide relatively accurate reward signals during the alignment phase.
>
>
> ---
>
> ` [Q1] Do you plan to open-source your dataset and pre-trained reward model? `
>
>
> Yes. We will open‑source our reward model, including the model weights as well as the code for training and evaluation. The dataset is subject to a stricter internal review; we are actively working through that process and will update it once approval is obtained.
>
>
> ---
>
> ` [Q2] Challenges of Real-Data Training in DPO/GRPO Settings? `
>
> Yes, we have explored the use of real videos in DPO training, for example, by using real videos as the *chosen* samples and generated videos as the *rejected* ones. However, this setup often leads to undesirable effects, such as over-sharpened outputs and the emergence of grid-like patterns in generated frames. We attribute this to the significant distributional gap between real and generated videos produced by current T2V models. Such a mismatch may result in suboptimal alignment outcomes.
>
> To mitigate this, possible solutions include (1) incorporating an auxiliary supervised fine-tuning (sft) loss to stabilize training and (2) increasing the $\beta$ coefficient in the DPO loss to enforce stronger KL regularization. Additionally, if the primary failure is degraded visual quality (assuming DPO focuses on aligning other aspects such as motion naturalness), we have verified that a short sft stage (around 50 steps) on high-quality real videos after DPO training can effectively restore visual fidelity. In summary, **if DPO leads to noticeable degradation in visual quality, a subsequent SFT stage on high-quality data can recover fidelity with very few steps, while still preserving the alignment benefits of DPO**.

---

### Official Review · Reviewer_Y3iN · 2025-06-29

**Clarity:** 2
**Significance:** 2
**Originality:** 3
**Rating:** 4
**Confidence:** 3

**Summary:**

This work introduces VideoReward, a three-dimensional reward model for video generation. To evaluate its effectiveness, the authors further propose three strategies: two training-time strategies (Flow-DPO and Flow-RWR) and one test-time strategy (Flow-NRG), all tailored for flow-based video generation models. Experimental results demonstrate the effectiveness of VideoReward and the proposed strategies on their internal VDM.

**Questions:**

Please refer to the weaknesses for detailed concerns.

**Ethical Concerns:**

["NO or VERY MINOR ethics concerns only"]

**Final Justification:**

The authors' response has successfully addressed my concerns. Thus, I raise my rating accordingly.

**Limitations:**

Yes.

**Paper Formatting Concerns:**

No or very minor formatting issues

**Quality:**

2

**Strengths And Weaknesses:**

**Strenth:**
1. Proposes three novel optimization strategies for flow-based models, covering both training and test-time stages.
2. Contributions in building a preference dataset, reward model, and reward benchmark, showcasing significant workload and effort.


**Weaknesses:**
1. **Unclear Motivation:** The motivation for introducing three strategies for flow-based models within a reward model is not clearly articulated. It would be helpful to clarify why these strategies are necessary.
2. **Limited Preference Aspects:** The human annotations only consider three preference aspects (VQ, MQ, and TA), which seem insufficient for evaluating video generation. Additional aspects, such as temporal coherence, should be considered, or the authors could provide further justification for focusing solely on these three aspects.
3. **Triplets Construction:** According to Line #99, preference video pairs are selected from distinct VDMs. However, due to the inherent quality differences between VDMs, selecting video pairs from the same VDM (with different random seeds) might result in more comparable positive and negative videos, enabling finer-grained preference modeling.
4. **Video Alignment Evaluation:** The evaluation is conducted exclusively on the authors’ internal pretrained model, which is insufficient to validate the generalizability of both the reward model and the proposed optimization strategies. Broader evaluations on more base models are necessary.
5. **Reference or Experimental Support:** Some claims lack adequate references or experimental support. For instance, Line #27 regarding previous reward models and Line #251 regarding the effects of full fine-tuning vs. LoRA fine-tuning require more evidence or citations.

---

> ### Author Rebuttal · Authors · 2025-07-31
>
> We sincerely appreciate your insightful comments and efforts in reviewing our manuscript. We respond to each of your comments one by one in what follows.
>
>
> ---
>
> ` [W1] Clarify the motivation for introducing three strategies for flow-based models within a reward model.`
>
> Thank you for your insightful comments. The three alignment algorithms are not isolated; they **all stem from a unified reinforcement learning framework** that aims to maximize reward with KL regularization. In other words, these strategies are three different implementations of the same optimization objective.
>
> These strategies were introduced to address the different practical needs in applying human preference alignment to flow-based models. Specifically, Flow-DPO and Flow-RWR are designed for training time, with our results showing Flow-DPO to be more effective. Flow-NRG, on the other hand, is designed for inference time scenarios where retraining is impractical.
>
> ---
>
> ` [W2] Limited Preference Aspects: Additional aspects, such as temporal coherence, should be considered, or the authors could provide further justification for focusing solely on these three aspects. `
>
>
> In our paper, we primarily focus on three aspects: Visual Quality (VQ), Motion Quality (MQ), and Text Alignment (TA). Section F provides a detailed explanation of the specific factors considered under each aspect. **Regarding temporal coherence, we treat it as a component of Motion Quality** (see Table 7). This categorization is inspired by the design of image reward models like ImageReward [1], which typically consider VQ and TA. Building on this, we introduce MQ to account for the temporal dimension of video preferences. We believe this three-aspect formulation is reasonable, and we agree that adopting a more fine-grained decomposition could be a valuable direction for future work.
>
> [1]. Imagereward: Learning and evaluating human preferences for text-to-image generation. NeurIPS 2023.
>
>
> ---
>
> ` [W3] Selecting video pairs from the same VDM (with different random seeds) might result in more comparable positive and negative videos, enabling finer-grained preference modeling `
>
> Thank you for the valuable suggestion. We agree that sampling pairs from the same VDM could yield finer-grained preferences. In our current setup, although the pairs are drawn from different VDMs, the models are comparable in capability, with similar mean MOS scores (see the table below). Here, MOS refers to the mean human opinion score, where annotators rate individual videos on a scale from 1 to 5. So this choice does not, in practice, prevent effective preference modeling. We acknowledge that intra‑model pairing is a promising direction and will explore it in future work.
>
>
> $$
> \begin{array}{l|ccc}
> \hline
> \textbf{Model} & \textbf{MOS VQ} & \textbf{MOS MQ} & \textbf{MOS TA} \newline
> \hline
> \text{Dreamina}  & 2.45 & 2.04 & 3.49 \newline
> \text{Luma}      & 2.23 & 2.11 & 3.45 \newline
> \text{Gen3}      & 2.42 & 2.16 & 3.26 \newline
> \text{Kling 1.0} & 2.34 & 2.20 & 3.53 \newline
> \text{Kling 1.5} & 2.37 & 2.22 & 3.54 \newline
> \hline
> \end{array}
> $$
>
>
> ---
>
> ` [W4] Broader evaluations on more base models are necessary. `
>
> As shown in Section C and Figure 7, our base model follows a standard VDM recipe—a 3D VAE coupled with a Transformer-based latent diffusion model. Most contemporary T2V models (e.g., Sora [2], Hunyuan Video [3], Wan2.1 [4]) adopt the same formulation. Additionally, our method is not tied to any architecture‑specific components. We believe that the results on this architecture are representative, and our approach can be applied to other T2V models without modification.
>
>
> [2]. Sora: Video generation models as world simulators.
> [3]. Hunyuanvideo: A systematic framework for large video generative models.
> [4]. Wan: Open and Advanced Large-Scale Video Generative Models.
>
> ---
>
> ` [W5-1] Reference or Experimental Support: Line #27 regarding previous reward models. `
>
> Previous reward models (e.g., VideoScore [5], LIFT [7]) were trained on datasets consisting primarily of low resolutions (often $\le$ 480p) and short durations (2s), whereas ours cover recent T2V models (most $\ge$ 720p) with higher resolutions and longer clips (3s -6s). The statistics are listed below. We will add these statistics in the revision.
>
> $$
> \begin{array}{lcc}
> \hline
> \textbf{Dataset} & \textbf{Resolution} & \textbf{Duration} \newline
> \hline
> \textit{Existing Preference Datasets} & & \newline
> \text{VideoFeedback [5]} & 256\times256\ - \ 576\times1024 & \text{1s}\ -\ \text{3s}\newline
> \text{VideoDPO [6]}      & 320\times512                          & \text{2s} \newline
> \text{LiFT-HRA [7]}      & 480\times720                          & \text{6s} \newline
> \hline
> \textit{Our Datasets} & & \newline
> \text{Ours}             & 384\times672\ - \ 768\times1408 & \text{3s}\ -\ \text{6s} \newline
> \hline
> \end{array}
> $$
>
> Furthermore, as shown in Table 2, while prior reward models perform well on GenAI-Bench, they perform significantly worse on VideoGen-RewardBench compared to ours. GenAI-Bench videos are shorter (2–2.5 seconds) and lower in quality, while VideoGen-RewardBench features longer clips (4–6 seconds) and higher resolution generated by modern video models.
>
> [5]. Videoscore: Building automatic metrics to simulate fine-grained human feedback for video generation. EMNLP 2024.
> [6]. Videodpo: Omni-preference alignment for video diffusion generation. CVPR 2025.
> [7]. LiFT: Leveraging Human Feedback for Text-to-Video Model Alignment. arXiv 2024.
>
> ---
>
> ` [W5-2] Reference or Experimental Support: Line #251 regarding the effects of full fine-tuning vs. LoRA fine-tuning `
>
> Thank you for your helpful suggestion. Regarding line #251 ("full-parameter updates were observed to hurt quality or even collapse the model"), our preliminary ablation shows that full-parameter fine-tuning more often degrades overall quality and is more susceptible to reward hacking issues than LoRA-based tuning. Recent works like SD3 [8] also favor LoRA over full fine-tuning in their post-training stage. We will add the supporting evidence in the revision.
>
>
> [8]. Scaling rectified flow transformers for high-resolution image synthesis. ICML 2024

---

> > ### Comment · Reviewer_Y3iN · 2025-08-06
> > **RE: Rebuttal**
> >
> > Thank you to the authors for the response, which has successfully addressed my concerns. I will raise my rating.

---

> > > ### Author Response · Authors · 2025-08-06
> > >
> > > Thank you so much for your quick response and for helping improve our work. We will incorporate your suggestions into the final version.

---

### Official Review · Reviewer_B8CS · 2025-06-30

**Clarity:** 3
**Significance:** 3
**Originality:** 3
**Rating:** 3
**Confidence:** 4

**Summary:**

This paper introduces an approach to enhance video generation by integrating human feedback, aiming to overcome issues such as unsmooth motion and misalignment between video content and prompts. The authors construct a comprehensive human preference dataset, focusing on advanced video generation models and incorporating pairwise annotations across multiple dimensions. The paper presents three alignment algorithms tailored for flow-based models: Flow-DPO, Flow-RWR, and an inference-time method, Flow-NRG.
Experimental results demonstrate that VideoReward outperforms existing reward models and that Flow-DPO achieves superior performance over Flow-RWR and conventional supervised fine-tuning methods. Additionally, Flow-NRG offers users the flexibility to assign custom weights to multiple objectives during inference.

**Questions:**

I will consider whether to adjust my score based on how my concerns are addressed and the comments of other reviewers.

**Ethical Concerns:**

["NO or VERY MINOR ethics concerns only"]

**Final Justification:**

The rebuttal solved most of my concerns. However, the technical novelty and theoretical explainability of the work remain unresolved. Therefore, my final score is borderline reject.

**Limitations:**

yes

**Quality:**

3

**Strengths And Weaknesses:**

Strengths:
1. The paper is well-written with clear motivation, addressing two key challenges: the quality of reward data in previous methods and their focus on non-Flow Matching paradigms.
2. The paper conducts extensive experiments, thoroughly verifying the performance of the proposed method, leading to reliable experimental conclusions.
3. The design of Noisy Reward Guidance to replace classifier-free guidance is intriguing.

Weaknesses:
1. In the Discussion on βt section, the paper ultimately opts for a time-independent constant β. There seems to be no significant difference between Equations 6 and 4, as they merely change the prediction target from diffusion noise to the FM velocity field. Consequently, is there a substantial distinction between Flow-DPO and Diffusion-DPO? Is it necessary to differentiate them? Similarly, for Flow-RWR, Equations 9 and 8 present similar issues, raising questions about the necessity of distinguishing between Flow-RWR and diffusion-based RWR.
2. The novelty of the paper is relatively limited. Although the introduced Noisy Reward Guidance is interesting, the core design concepts, such as Flow-DPO, do not seem to represent significant improvement or innovation over previous work.
3. I am curious about the performance difference of the proposed method when using the original classifier-free guidance instead of Noisy Reward Guidance.
4. According to the quantitative results on VBench, there does not appear to be a significant improvement in the total metric compared to the pre-trained model.

Minor Weaknesses:
1. The Efficient Reward on Noisy Latents section lacks detailed information, which can cause confusion. For instance, the statement "We then adopt the Bradley–Terry loss to learn the reward function from these noised videos" is unclear whether the loss training is based on noisy latents or pixel space noisy videos.

---

> ### Author Rebuttal · Authors · 2025-07-31
>
> We sincerely appreciate your insightful comments and efforts in reviewing our manuscript. We respond to each of your comments one by one in what follows.
>
> ---
>
> ` [W1] Is it necessary to differentiate between Diffusion-DPO and Flow-DPO?`
>
>
> Thanks for asking this clarification question! A straightforward way to extend Diffusion-DPO to flow matching is indeed to replace the diffusion's noise prediction term with the FM's velocity prediction term. However, **this heuristic lacks theoretical grounding, so whether it works is unpredictable**.
>
> We first utilize the connection between diffusion and flow matching and derive a vanilla Flow-DPO objective that naturally carries a timestep-dependent $\beta_t$. Our analysis shows that this form induces a timestep-dependent KL constraint, which biases alignment towards high-noise (early) steps and triggers noticeable reward hacking on certain dimensions. Motivated by the DDPM simplification from Eq.12 to Eq.14, we then test a variant that replaces $\beta_t$ with a time-independent constant $\beta$; this version performs better overall and we term it as "Flow-DPO".
>
> Thus, although Flow‑DPO looks similar to Diffusion‑DPO in form, **we provide the theoretical rationale and empirical evidence for why the constant‑$\beta$ variant is preferable to the timestep‑dependent one**. Throughout the paper, we stress that **Flow‑DPO is an extension of Diffusion‑DPO to flow‑matching models**. Given that leading image generation models (e.g., SD3 [1], FLUX [2]) and video generation models (e.g., Sora [3], Wan2.1 [4], Seedance [5]) now adopt flow matching in place of traditional diffusion, we believe this distinction is necessary to keep the theory reasonable and making the framework more intuitive for the community.
>
> [1]. Scaling Rectified Flow Transformers for High-Resolution Image Synthesis.
> [2]. FLUX.
> [3]. Sora: Video generation models as world simulators.
> [4]. Wan: Open and Advanced Large-Scale Video Generative Models.
> [5]. Seedance 1.0: Exploring the Boundaries of Video Generation Models.
>
> ---
>
>
> ` [W2] The novelty of the paper is relatively limited.`
>
> We respectfully disagree with the assessment that the novelty of our work is limited. While it is true that some components were built upon existing paradigms in other fields, we emphasize that human preference alignment in video generation remains in its early stage, with several initial challenges yet to be adequately addressed. These include: (1) the mismatch between existing preference datasets and modern T2V models; (2) the underutilized potential of VLM-based reward modeling; and (3) the limited applicability of alignment algorithms like DPO. **Our work aims to propose a comprehensive alignment pipeline tailored to flow-based T2V models and validate its effectiveness in improving video generation**.
>
> Importantly, our contributions go beyond straightforward adaptations. For example, we show that directly extending Diffusion-DPO to flow-based models by leveraging the connection between diffusion and flow matching leads to reward hacking and suboptimal performance. Our Flow-DPO addresses this with a constant KL constraint, resulting in more stable and effective alignment. While Flow-DPO is similar to Diffusion-DPO in form, **we provide both theoretical rationale and empirical evidence to support its effectiveness.** Likewise, our proposed Flow-NRG offers a novel inference-time method for distribution alignment without additional training. **Beyond the alignment algorithms, our work also contributes insights into dataset construction and reward model design**, offering a comprehensive pipeline to preference alignment for video generation.
>
>
> ---
>
> ` [W3] The performance difference between CFG and Flow-NRG. `
>
> Thank you for your question, and we apologize for the confusion. Noisy Reward Guidance and classifier-free guidance (CFG) are orthogonal techniques. In all our experiments, the model first uses CFG to obtain an initial velocity, followed by Noisy Reward Guidance to refine it.
>
> As shown in Table 5, we directly compare the full method (CFG + Noisy Reward Guidance) with the CFG-only baseline. The reported win rates (%) reflect the performance gains brought by Noisy Reward Guidance on top of CFG. We will clarify this point more explicitly in the revision to avoid further confusion.
>
>
> ---
>
> ` [W4] VBench results show no significant overall improvement over the pre-trained model. `
>
>
> While the VBench score shows a modest increase from 83.19 to 83.41 after applying DPO (Table 3), this improvement is more significant than it appears. We argue that in generative AI, relying solely on automatic metrics to assess model capability remains challenging and can mask perceptual gains. **Beyond VBench, our user study (Fig. 5) shows 44.0% wins for the DPO‑tuned model versus 27.0% for the pretrained baseline (29.0% ties).** These consistent advantages indicate that DPO significantly improves visual quality and alignment despite the modest increase in the VBench score. We encourage the reviewer to view the video examples in the supplementary materials for a visual comparison of the improved generation quality.
>
> ---
>
> ` [Q1] Clarify reward training on noisy latents or pixel space`
>
> We apologize for the confusion caused by the lack of detail. For the "Efficient Reward on Noisy Latents" section, the loss is computed on noisy latents. As noted in line 208, we train a time-dependent reward model $r_\theta(·, t)$ directly in **latent space** to avoid the costly backpropagation through the full VAE decoder required for computing ∇r in pixel space.

---

> > ### Comment · Reviewer_B8CS · 2025-08-05
> > **Thanks for the rebuttal**
> >
> > I sincerely appreciate the authors’ response, which has addressed some of my questions. Nevertheless, I still have the following concerns:
> > 1. The clarification regarding the Flow-DPO component did not fully convince me. Although the authors describe how Diffusion-DPO is extended to Flow-DPO, the key modification that adopts a time-independent constant β is motivated by a variant already discussed in the original DDPM paper. Hence, the core conceptual novelty appears limited. The same issue holds for Flow-RWR. I therefore question whether introducing the separate notions of Flow-DPO and Flow-RWR as contributions is warranted. In addition, existing techniques are also adopted in other contributions, such as using the Bradley-Terry model with ties in Reward Modeling.
> > 2. Although the rebuttal repeatedly emphasizes the "theoretical rationale," I do not see an in-depth theoretical analysis of Flow-DPO or Flow-RWR in the paper. Many conclusions rest on empirical observations rather than rigorous theoretical treatment, leaving limited theoretical support and interpretability.
> > 3. The gains on VBench are marginal. Despite leveraging a large dataset and multiple technical strategies, the performance improvements remain modest.
> >
> > In summary, while the amount of engineering effort, e.g., constructing a large-scale dataset, is commendable, the work still seems constrained in both technical novelty and performance benefit.

---

> > > ### Author Response · Authors · 2025-08-06
> > >
> > > Thanks for your continued engagement and for providing further detailed feedback. We respond to each of your comments one by one in what follows.
> > >
> > > ---
> > >
> > > ` [W1 & W2] The core conceptual novelty of Flow-DPO and Flow-RWR appears limited. In addition, existing techniques are also adopted in other contributions, such as using the Bradley-Terry model with ties in Reward Modeling. `
> > >
> > > The core contribution of our work is not to propose an entirely new algorithm, but to address a critical and urgent problem: **how to effectively and stably align modern, state-of-the-art flow-based video models with human preferences**.
> > >
> > > Before our work, the field lacked a high-quality preference dataset, and the design space of VLM-based reward models remained under-explored. We construct a large-scale human preference dataset focused on modern video generation models, incorporating pairwise annotations across multi-dimensions and examine how annotations and various design choices impact the video reward model's rewarding efficacy. With leading video models (e.g., Wan2.1, Sora, Hunyuan Video) now rapidly adopting flow matching, our work demonstrates how existing alignment algorithms can be adapted to this paradigm and validates their effectiveness. Additionally, we introduce Flow-NRG for inference-time use when retraining is infeasible.
> > >
> > > In summary, our empirical findings offer practical design insights and strong evidence that human feedback can significantly boost the performance of state-of-the-art video generation models. As pretraining for video models stabilizes, our work highlights a promising new path to improve video quality and opens up new opportunities for future research in this area.
> > >
> > > ---
> > >
> > > ` [W3] The gains on VBench are marginal.`
> > >
> > > We agree that the numerical improvement on VBench is modest.
> > >
> > > However, we would like to respectfully argue that VBench is not always a definitive indicator of generative video quality. A notable example from the official VBench leaderboard is that the smaller Wan2.1-T2V-1.3B model scores higher than the Wan2.1-T2V-14B model, despite the community's consensus that the 14B model is significantly superior.
> > > $$
> > > \begin{array}{c|ccc}
> > > \hline
> > > \text{Model Name} & \text{Total Score} & \text{Quality Score} & \text{Semantic Score} \newline
> > > \hline
> > > Wan2.1-T2V-1.3B  & 84.26 & 85.30 & 80.09       \newline
> > > Wan2.1-T2V-14B  & 83.69 & 85.59 & 76.11       \newline
> > > \hline
> > > \end{array}
> > > $$
> > >
> > > We believe a more accurate assessment of our method's impact is provided by our human evaluation and win rate via VideoReward. We would kindly direct the reviewer to:
> > >
> > > 1.  **The User Study (Figure 5):** This large-scale study, conducted on 400 diverse prompts from VideoGen-Eval and judged by multiple annotators, shows a clear and significant human preference for our aligned model (44% vs. 27%).
> > >
> > > 2.  **The Reward Model Win Rate (Table 3):**  This metric directly measures the success of our DPO alignment. The high win rate demonstrates that our aligned model is consistently judged as superior to the pretrained model by the VideoReward model.

---

> ### Comment · Reviewer_B8CS · 2025-08-07
> **Thanks for the discussion**
>
> Thanks for the further discussion. It has resolved my concerns about the limited performance gains. However, although the authors state that the goal is to "align modern, state-of-the-art flow-based video models with human preferences," the technical novelty and theoretical explainability of the work are still limited. Considering both the strengths and weaknesses of the paper, I will maintain my original score and leave the final decision to the ACs.

---

### Official Review · Reviewer_sgc3 · 2025-07-04

**Clarity:** 4
**Significance:** 4
**Originality:** 3
**Rating:** 5
**Confidence:** 4

**Summary:**

The paper provides a complete solution for improving the visual quality of flow-based video generation models through human feedback. This includes:
* A large-scale multi-dimensional human preference dataset,
* A video reward model trained on the dataset,
* A reward model benchmark dataset containing 26.5K video pairs,
* 3 algorithms for improving video generation with the help of the reward model.
Each individual component contain their own innovations and empirical justifications:
The preference dataset is collected from more recent video generators to raise the standard, and includes both point-wise and pair-wise annotations.
Next, several design decisions for reward modeling are explored, with the Bradley-Terry model with ties achieving the best performance.
Finally, the reward model is used to improve a flow-based video generator using newly-derived Flow-DPO, Flow-RWR, and Flow-NRG formulations. Flow-DPO improves the generation over the base model both quantitatively and visually.

**Questions:**

* How does Flow-NRG. compare with other methods?
* The paper mentions reward hacking issues with Flow-DPO when using time-dependent $\beta$. Could the authors elaborate how this reward hacking manifests visually in the generated videos?

**Ethical Concerns:**

["NO or VERY MINOR ethics concerns only"]

**Final Justification:**

I maintain my rating of accept. The paper is well-written with a lot of useful detail, and I have no major concern over the validity of the methods.

**Limitations:**

yes

**Quality:**

4

**Strengths And Weaknesses:**

### Strengths
* This paper provides a complete guideline for building a full RLHF workflow for flow-based video generation models and more. It serves as a valuable resource for future research.
* The paper includes comprehensive ablations, such as the comparison between point-wise and pair-wise rewards, as well as the scaling factors for Flow-DPO. They are good references for future works.
* The proposed reward model achieves SOTA performance, while the video generator show promising improvement after DPO.
* The paper is well written and logically well-structured, including just the right amount of detail for such an involved pipeline without overwhelming the reader. For example,

### Weaknesses
* The quantitative evaluation for Flow-NRG seems to be missing.

---

> ### Author Rebuttal · Authors · 2025-07-31
>
> We sincerely appreciate your insightful comments and efforts in reviewing our manuscript. We respond to each of your comments one by one in what follows.
>
> ---
>
> ` [W1 & Q1] The quantitative evaluation for Flow-NRG. `
>
> Table 5 in the paper reports Flow‑NRG’s quantitative evaluation (via quantitative) on VideoGen‑Eval under custom reward weights. For convenience, the numbers are reproduced below. The three columns (VQ, MQ, TA) are the win rates (%) against the vanilla model when we steer the generator with different reward-weight triplets $w_{\text{VQ}}:w_{\text{MQ}}:w_{\text{TA}}$, where $r=w_{\text{VQ}}*r_{\text{VQ}}+w_{\text{MQ}}*r_{\text{MQ}}+w_{\text{TA}}*r_{\text{TA}}$.
>
> $$
> \begin{array}{c|ccc}
> \hline
> \text{$w_{\text{VQ}}:w_{\text{MQ}}:w_{\text{TA}}$} & \text{VQ} & \text{MQ} & \text{TA} \newline
> \hline
> 0.0:0.0:1.0  & 60.56 & 46.48 & 70.42       \newline
> 0.1:0.1:0.8  & 66.50 & 63.73 & 60.86       \newline
> 0.1:0.1:0.6  & 68.94 & 67.59 & 53.28       \newline
> 0.5:0.5:0.0  & 86.43 & 93.23 & 26.65       \newline
> \hline
> \end{array}
> $$
>
>
> The results show Flow‑NRG reliably steers generation toward the weighted objectives and improves alignment. Compared with other methods in Table 1, Flow‑NRG achieves performance comparable to SFT and somewhat below Flow‑DPO, which is reasonable given that Flow‑NRG is purely inference‑time.
>
>
> ---
>
>
> ` [Q2] The visual manifestations of reward hacking`
>
> Thank you for the suggestion! We observe that Flow-DPO with time-dependent $\beta$ is more prone to hacking issues, **including abnormal color shifts, unnatural brightness and contrast, and reduced diversity** (e.g., the model tends to enlarge the main subject to avoid penalties from missing local details). These artifacts degrade the general quality yet still yield higher rewards. We will add qualitative examples of these hacking issues in the revision.

---

> > ### Author Response · Authors · 2025-08-08
> >
> > We sincerely thank Reviewer sgc3 for your thoughtful and detailed review. We are very pleased that you recognized that our work **provides a complete guideline for building a full RLHF workflow for flow-based video generation models and serves as a valuable resource for future research**.
> >
> > Your comments regarding the quantitative evaluation of Flow-NRG and the visual manifestations of reward hacking in generated videos have been extremely valuable in helping us refine our work. To that end, we have provided detailed responses in our rebuttal to address your concerns.
> >
> > As the discussion phase draws to a close, we would be grateful to know whether our responses have resolved your initial concerns. We greatly appreciate your time and would welcome any further feedback.

---

> > > ### Comment · Reviewer_sgc3 · 2025-08-08
> > >
> > > Thank you for the response. I totally missed Table 5, and thanks for sharing the reward hacking symptoms. This resolves all of my concerns!

---

> > > > ### Author Response · Authors · 2025-08-08
> > > >
> > > > Thanks for your response! We are glad to know your concerns are addressed.

---

### Note · Authors · 2025-08-12

Dear Reviewers and Area Chairs,

We sincerely thank all reviewers for the thoughtful feedback and constructive discussion. We are encouraged that the reviewers have either raised their scores or maintained current ratings. Reviewer sgc3 recognized that our work provides a complete guideline for building a full RLHF workflow for flow-based video generation models and serves as a valuable resource for future research; Reviewer B8CS found the design of Noisy Reward Guidance intriguing; Reviewer Y3iN appreciated the novelty of our strategies and acknowledged the significant workload and effort behind our work; and Reviewer zALS affirmed the effectiveness of our algorithms. We greatly appreciate the constructive spirit throughout the discussion.

In our rebuttal, we have carefully addressed every reviewer's question in detail, with particular attention to: (1) the technical details of Flow-NRG; (2) the motivation for introducing three alignment algorithms for flow-based models; (3) the VBench performance gains; and (4) the design of the preference dataset.  We are grateful for all feedback, which has improved the paper’s clarity and strengthened our arguments.

Regarding contributions, we emphasize that human preference alignment for video generation remains in its early stage, with several initial challenges yet to be adequately addressed. Our work targets a central, urgent question: **how to effectively and stably align modern, state-of-the-art flow-based video models with human preferences.** We contribute along three fronts: dataset construction, reward model design, and the alignment algorithms, culminating in a comprehensive pipeline for preference alignment in video generation. As pretraining for video models stabilizes, our work highlights a promising new path to improve video quality and opens up new opportunities for future research in this area.

We sincerely hope that our responses and new evidence convey both the insight and the potential impact of our paper, and we are grateful for the reviewers' valuable feedback that guided these improvements. Thank you again for your time and thoughtful consideration.

Best regards,

Authors of paper #15988

---

### Decision · Program_Chairs · 2025-09-17

**Decision:**

Accept (poster)

**Comment:**

This paper proposed a complete solution of improving video generation with human feedback, including a large-scale multi-dimensional human preference dataset, a video reward model and video reward benchmark data set, and three methods (Flow-DPO, Flow-RWR, and Flow-NRG) to improve video generation model.

Advantages pointed out by reviewers are 1) multiple contribution of a full RLHF workflow including preference data set, video reward model, 3 methods to enhance video generation  2)comprehensive experiments with ablation studies, showing the promising results for the proposed reward model 3) Some novel design like noisy reward guidance to replace classifier-free guidance.

After rebuttal, the major concern remained is methodological novelty, i.e., Flow-DPO and Flow-RWR are direct adaptations of Diffusion-DPO and RWR to flow matching. However, consider the contributions of other aspects and the advantages, I will recommend "Accept (poster)".